# Gene Regulatory Network Inference in the Presence of Dropouts: a Causal View

**Haoyue Dai**[1]  **Ignavier Ng**[1]  **Gongxu Luo**[2]  **Peter Spirtes**[1]  **Petar Stojanov**[3]  **Kun Zhang**[1,2]
[1]Department of Philosophy, Carnegie Mellon University
[2]Machine Learning Department, Mohamed bin Zayed University of Artificial Intelligence
[3]Cancer Program, Eric and Wendy Schmidt Center, Broad Institute of MIT and Harvard

## Abstract

Gene regulatory network inference (GRNI) is a challenging problem, particularly owing to the presence of zeros in single-cell RNA sequencing data: some are biological zeros representing no gene expression, while some others are technical zeros arising from the sequencing procedure (aka dropouts), which may bias GRNI by distorting the joint distribution of the measured gene expressions. Existing approaches typically handle dropout error via imputation, which may introduce spurious relations as the true joint distribution is generally unidentifiable. To tackle this issue, we introduce a causal graphical model to characterize the dropout mechanism, namely, *Causal Dropout Model*. We provide a simple yet effective theoretical result: interestingly, the conditional independence (CI) relations in the data with dropouts, after deleting the samples with zero values (regardless if technical or not) for the conditioned variables, are asymptotically identical to the CI relations in the original data without dropouts. This particular test-wise deletion procedure, in which we perform CI tests on the samples without zeros for the conditioned variables, can be seamlessly integrated with existing structure learning approaches including constraint-based and greedy score-based methods, thus giving rise to a principled framework for GRNI in the presence of dropouts. We further show that the causal dropout model can be validated from data, and many existing statistical models to handle dropouts fit into our model as specific parametric instances. Empirical evaluation on synthetic, curated, and real-world experimental transcriptomic data comprehensively demonstrate the efficacy of our method.

## 1 Introduction

Gene regulatory networks (GRNs) represent the causal relationships governing gene activities in cells (Levine & Davidson, 2005), essential for understanding biological processes and diseases like cancer (Ito et al., 2021; Huang et al., 2020; Parrish et al., 2021). Typically, for $p$ genes, the GRN is a graph consisting of nodes $\mathbf{Z} = \{Z_i\}_{i=1}^p$ representing gene expressions, and directed edges representing cross-gene regulations. Traditional lab-based GRNI involves gene knockout experiments, but conducting all combinatorial interventions is challenging. In contrast, observational expression data is abundant by RNA-sequencing. In the past decade, single-cell RNA-sequencing (*scRNA-seq*) has become prevalent, enabling comprehensive studies in cancer Neftel et al. (2019); Boiarsky et al. (2022) and genomic atlases Regev et al. (2017); Consortium et al. (2018; 2022) at the individual cell level. Causal discovery techniques for GRNI have also been steadily developed to leverage these advances (Wang et al., 2017; Zhang et al., 2017; Belyaeva et al., 2021; Zhang et al., 2021; 2023).

Despite the advantage of scRNA-seq, a fundamental challenge known as the *dropout* issue arises. scRNA-seq data is known to exhibit an abundance of zeros. While some of these zeros correspond to genuine biological absence of gene expression (Zappia et al., 2017; Alberts et al., 2002), some others are technical zeros arising from the sequencing procedure, commonly referred to as *dropouts* (Jiang et al., 2022; Ding et al., 2020). Various factors are commonly acknowledged to contribute to the occurrence of dropouts, including low RNA capture efficiency (Silverman et al., 2020; Jiang et al., 2022; Kim et al., 2020; Hicks et al., 2018), intermittent degradation of mRNA molecules (Pierson & Yau, 2015; Kharchenko et al., 2014), and PCR amplification biases (Fu et al., 2018; Tung et al., 2017). The dropout issue has been shown to introduce biases and pose a threat to various downstream tasks, including gene regulatory network inference (Jiang et al., 2022; Van Dijk et al., 2018).

Dealing with dropouts in scRNA-seq data has been approached through two main strategies. One approach involves using probabilistic models such as zero-inflated models (Pierson & Yau, 2015; Kharchenko et al., 2014; Saeed et al., 2020; Yu et al., 2023; Min & Agresti, 2005; Li & Li, 2018; Tang et al., 2020) or hurdle models (Finak et al., 2015; Qiao et al., 2023) to characterize the distribution of gene expressions with dropouts. However, these methods may have limited flexibility due to the restrictive parametric assumptions (Svensson et al., 2018; Kim et al., 2020). Another approach is imputation (Van Dijk et al., 2018; Li & Li, 2018; Huang et al., 2018; Amodio et al., 2019; Lopez et al., 2018), where all zeros are treated as missing values and imputed to estimate the underlying distribution of genes without dropouts. However, imputation methods often lack a theoretical guarantee due to the inherent unidentifiability of the underlying distribution. Empirical studies also demonstrate mixed or no improvement when using imputation on various downstream tasks (Hou et al., 2020; Jiang et al., 2022). Overall, despite various attempts, there is currently still not a principled and systematic approach to effectively address the dropout issue in scRNA-seq data.

While GRNs represent the *causal* regulations among genes in the expression process, can we also extend this understanding to the *causal* mechanisms of dropouts in the sequencing process? With this motivation, we abstract the common understanding of dropout mechanisms by proposing a causal graphical model, called *causal dropout model* (§2.2). We then recognize that the observed zeros in scRNA-seq data resulting from dropouts are *non-ignorable*, implying that the distribution of the original data is irrecoverable without further assumptions (§2.4). However, surprisingly, we show theoretically that, given such qualitative understanding of dropout mechanism, we could simply *ignore* the data points in which the conditioned variables have zero values, leading to consistent estimation of conditional independence (CI) relations with those in the original data (§2.5). This insight then readily bridges the gap between dropout-tainted measurements and GRNI with an asymptotic correctness guarantee (§3). We further provide a systematic way to verify such dropout mechanisms from observations (§4).

**Contributions.** **1)** Theoretically, the proposed *causal dropout model* is, to the best of our knowledge, the first theoretical treatment of the dropout issue in a fully non-parametric setting. Most existing parametric models fit into our framework as specific instances. **2)** Empirically, extensive experimental results on synthetic, curated, and real-world experimental transcriptomic data with both traditional causal discovery algorithms and GRNI-specific algorithms demonstrate the efficacy of our method.

## 2 Causal Model for Dropouts and Motivation of our Approach

In this section, we develop a principled framework to model the dropout mechanisms in scRNA-seq data, and describe the motivation of our proposed GNRI approach that will be formally introduced in §3. We first introduce a causal graphical model to characterize the dropout mechanism in §2.2, and demonstrate in §2.3 how existing statistical models commonly used to handle dropouts fit into our framework as specific parametric instances. We then discuss in §2.4 the potential limitations of imputation methods, demonstrating that the underlying distribution without dropouts is generally unidentifiable. Lastly, we demonstrate in §2.5 that, despite the unidentifiability of the distribution, the entailed conditional (in)dependencies can be accurately estimated, which serves as the key motivation of our proposed GNRI approach in §3 (grounded on the causal graphical model presented in §2.2).

### 2.1 Warmup: Causal Graphical Models and Causal Discovery

Causal graphical models use directed acyclic graphs (DAGs) to represent causal relationships (directed edges) between random variables (vertices) (Pearl, 2009; Spirtes et al., 2000). Causal discovery aims to identify the causal graph from observational data. In this paper, by "observational" we assume the data samples are drawn i.i.d., without interventions or heterogeneity. In the context of GRNI, we focus on cells produced in a same scRNA-seq run without different perturbations (Dixit et al., 2016).

A typical kind of causal discovery methods is called *constraint-based* methods, e.g., the PC algorithm (Spirtes et al., 2000). These methods associate the conditional (in)dependencies $(X \perp\!\!\!\perp Y | \mathbf{S})$ in data with the (un)connectedness in graph, characterized by the d-separation patterns $(X \perp\!\!\!\perp_d Y | \mathbf{S})$, and accordingly recover the underlying causal graph structure, up to a set of equivalent graphs known as Markov equivalence classes (MECs), represented by complete partial DAGs (CPDAGs). Note that as CI tests do not require any specific assumptions about the underlying distributions of the variables (Rosenbaum, 2002; Zhang et al., 2011), constraint-based methods can in theory work nonparametrically for arbitrary distributions consistent with the graph.

## 2.2 CAUSAL DROPOUT MODEL: A CAUSAL GRAPHICAL MODEL FOR DROPOUTS

Let $G$ be a DAG with vertices partitioned into four sets, $\mathbf{Z} \cup \mathbf{X} \cup \mathbf{D} \cup \mathbf{R}$. Denote by $\mathbf{Z} = \{Z_i\}_{i=1}^p$ the true underlying expression variables of the $p$ genes. The subgraph of $G$ on $\mathbf{Z}$ is the gene regulatory network (GRN) that we aim to recover from data. However, we do not have full access to true expressions $\mathbf{Z}$, but only the sequenced observations $\mathbf{X} = \{X_i\}_{i=1}^p$ contaminated by dropout error. We use boolean variables $\mathbf{D} = \{D_i\}_{i=1}^p$ to indicate for each gene whether technical dropout happens in sequenced individual cells: $D_i = 1$ indicates dropout and $0$ otherwise. Hence $\mathbf{X}$ are generated by:

$$X_i = (1 - D_i) * Z_i. \tag{1}$$

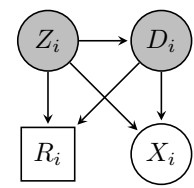

Similar to $\mathbf{Z}$, we also do not have full access to $\mathbf{D}$ in practice (shown as the gray color in Figure 1): whether a zero is biological or technical is still unknown to us. To help address such issue, we further introduce zero observational indicators $\mathbf{R} = \{R_i\}_{i=1}^p$ defined as $R_i = D_i$ OR $\mathbb{1}(Z_i = 0)$, i.e., zeros are either technical or biological. By definition in Equation (1), $R_i$ can be fully read from $X_i$ by $R_i = \mathbb{1}(X_i = 0)$. For any subset $\mathbf{S} \subset [p] := \{1, \ldots, p\}$, we denote the corresponding random vector by e.g., $\mathbf{X_S} = \{X_i : i \in \mathbf{S}\}$. Specifically, for boolean indicator variables, we use e.g., $\mathbf{R_S} = \mathbf{1}$ to denote $\mathrm{OR}_{i \in \mathbf{S}} R_i = 1$, and $\mathbf{R_S} = \mathbf{0}$ for $\mathrm{AND}_{i \in \mathbf{S}} R_i = 0$. We call such a proposed causal graphical model for modelling dropouts a *Causal Dropout Model* (CDM) throughout this paper.

Figure 1: Causal graph for dropouts. Gray nodes are underlying partially observed variables and white nodes are observed ones.

Note that in the causal graph depicted in Figure 1, the node $R_i$ is represented by a square, while the others are represented by circles. This distinction is made because $R_i$ serves as an auxiliary indicator instead of a variable with an atomic physical interpretation. The directed edges into $X_i$ and $R_i$ from $Z_i$ and $D_i$ are deterministic, i.e., the causal model can be sufficiently represented by $Z_i$ and any one of $D_i$, $R_i$, or $X_i$, while the redundant inclusion of all four variables is merely for the convenience of derivation. The key component is the edge $Z_i \to D_i$, representing the *self-masking* dropout mechanism, i.e., whether a gene is undetected in a particular cell is determined by the gene's true expression level in the cell. This aligns with the several commonly acknowledged reasons of dropouts (Silverman et al., 2020; Jiang et al., 2022; Kim et al., 2020; Hicks et al., 2018). But note that such self-masking assumption can also be relaxed, allowing for edges $Z_j \to R_i$, as discussed in §4.2.

## 2.3 EXISTING MODELS AS SPECIFIC INSTANCES OF THE CAUSAL DROPOUT MODEL

In addition to the common biological understandings on the dropout mechanisms, many existing statistical models to handle dropouts also fit into our introduced framework as specific parametric instances. Below, We give a few illustrative examples (detailed analysis is available in Appendix B.1):

**Example 1** (Dropout with the fixed rates). $D_i \sim \text{Bernoulli}(p_i)$, i.e., the gene $Z_i$ gets dropped out with a fixed probability $p_i$ across all individual cells. The representative models in this category are the zero-inflated models (Pierson & Yau, 2015; Kharchenko et al., 2014; Saeed et al., 2020; Yu et al., 2023; Min & Agresti, 2005) and the hurdle models (Finak et al., 2015; Qiao et al., 2023). Biologically, this can be explained by the random sampling of transcripts during library preparation, regardless of the true expressions of genes. Graphically, in this case the edge $Z_i \to D_i$ is absent.

**Example 2** (Truncating low expressions to zero). $D_i = \mathbb{1}(Z_i < c_i)$, i.e., the gene $Z_i$ gets dropped out in cells whenever its expression is lower than a threshold $c_i$. A typical kind of such truncation models is, for simple statistical properties, the truncated Gaussian copula (Fan et al., 2017; Yoon et al., 2020; Chung et al., 2022). Biologically, such truncation thresholds $c_i$ (quantile masking (Jiang et al., 2022)) can be explained by limited sequencing depths. Graphically, in this case the edge $Z_i \to D_i$ exists.

**Example 3** (Dropout probabilistically determined by expressions). $D_i \sim \text{Bernoulli}(F_i(\beta_i Z_i + \alpha_i))$, where $\beta_i < 0$ and $F_i$ is monotonically increasing (typically as CDF of probit or logistic (Cragg, 1971; Liu, 2004; Miao et al., 2016)), i.e., a gene $Z_i$ may be detected (or not) in every cell, while the higher it is expressed in a cell, the less likely it gets dropped out. Biologically, this can be explained by inefficient amplification. Graphically, the edge $Z_i \to D_i$ also exists, and is non-deterministic.

As shown above, many existing statistical models to handle dropouts can fit into our proposed causal dropout graph as parametric instances. Furthermore, it is important to note that our causal model itself, as well as the corresponding causal discovery method proposed in §3, are nonparametric. Hence, our model is more flexible and robust in practice. As is verified in §5.1, even on data simulated according to these parametric models, our method outperforms the methods specifically designed for them.

## 2.4 POTENTIAL THEORETICAL ISSUES WITH THE IMPUTATION METHODS

While many parametric models exist (§2.3), imputation methods have become more prevalent to handle dropouts in practice (Hou et al., 2020; Van Dijk et al., 2018; Li & Li, 2018; Huang et al., 2018; Lopez et al., 2018). Imputation methods treat the excessive zeros as "missing holes" in the expression matrix and aim to fill them using nearby non-zero expressions. However, a fundamental question arises: is it theoretically possible to fill in these "missing holes"? In the context of the causal dropout model in Equation (1), we demonstrate that, in general, the answer is negative, as the underlying true distribution $p(\mathbf{Z})$ is unidentifiable from observations $p(\mathbf{X})$ (detailed analysis in Appendix B.2):

Firstly, according to the missing data literature, the underlying joint distribution $p(\mathbf{Z})$ is *irrecoverable* due to the *self-masking* dropout mechanism (Enders, 2022; Mohan et al., 2013; Shpitser, 2016):

**Example 4.** Consider the following two models consistent with Example 3 (Miao et al., 2016):

1. $Z_i \sim \mathrm{Exp}(2)$, $D_i \sim \mathrm{Bernoulli}(\mathrm{sigmoid}(\log 2 - Z_i))$.
2. $Z_i \sim \mathrm{Exp}(1)$, $D_i \sim \mathrm{Bernoulli}(\mathrm{sigmoid}(Z_i - \log 2))$.

With even the same parametric logistic dropout mechanism and different distributions of $Z_i$, the resulting observations $X_i$ share exactly the same distribution, as $2e^{-2z} \cdot \frac{1}{1+e^{\log 2 - z}} = e^{-z} \cdot \frac{1}{1+e^{z - \log 2}}$.

Secondly, different from missing data, the "missing" entries here cannot be precisely located, i.e., $\mathbf{D}$ variables are latent. Therefore, even under the simpler fixed-dropout-rate scheme as in Example 1 (dropout happens *completely-at-random* (CAR) (Little & Rubin, 2019)), $p(\mathbf{Z})$ remains irrecoverable:

**Example 5.** Consider the following two models consistent with Example 1:

1. $Z_i \sim \mathrm{Bernoulli}(0.5)$, $D_i \sim \mathrm{Bernoulli}(0.6)$.
2. $Z_i \sim \mathrm{Bernoulli}(0.8)$, $D_i \sim \mathrm{Bernoulli}(0.75)$.

With the CAR dropout mechanism and different $p(Z_i)$, the observed $X_i$ follow a same $\mathrm{Bernoulli}(0.2)$.

## 2.5 MOTIVATION: PRESERVATION OF CONDITIONAL (IN)DEPENDENCIES IN NON-ZERO PARTS

We have demonstrated the general impossibility of recovering the true underlying distribution $p(\mathbf{Z})$ from dropout-contaminated observations $\mathbf{X}$. However, for the purpose of GRNI, a structure learning task, it is not necessary to accurately recover $p(\mathbf{Z})$. Alternatively, the graph structure among $\mathbf{Z}$ can be sufficiently recovered by the CI relations entailed by $p(\mathbf{Z})$. Hence, the key question becomes whether the CIs of $p(\mathbf{Z})$ can survive the inestimable distortion introduced by dropouts and leave trace in $p(\mathbf{X})$.

**Example 6.** Consider the linear structural equation model (SEM) consistent with a chain structure $Z_1 \rightarrow Z_2 \rightarrow Z_3$:

$$Z_1 = E_1$$
$$Z_2 = Z_1 + E_2$$
$$Z_3 = 3Z_2 + E_3$$
$$E_1, E_2, E_3 \sim \mathcal{N}(0,1)$$

and the logistic dropout scheme: $D_i \sim \mathrm{Bernoulli}(\mathrm{logistic}(-2Z_i))$ for $i = 1, 2, 3$. The scatterplots (a) and (d) in Figure 2 demonstrate $Z_1 \not\perp\!\!\!\perp Z_3$ and $X_1 \not\perp\!\!\!\perp X_3$ marginally. Now condition on the middle variable. The d-separation $Z_1 \perp\!\!\!\perp_d Z_3 | Z_2$ implies that $Z_1 \perp\!\!\!\perp Z_3 | Z_2$, as supported by (b) and (c) with $Z_2 = 0, 1$ respectively. However, in contrast, one observes $X_1 \not\perp\!\!\!\perp X_3 | X_2$, as supported by the scatters in (e): with the concentration of points along

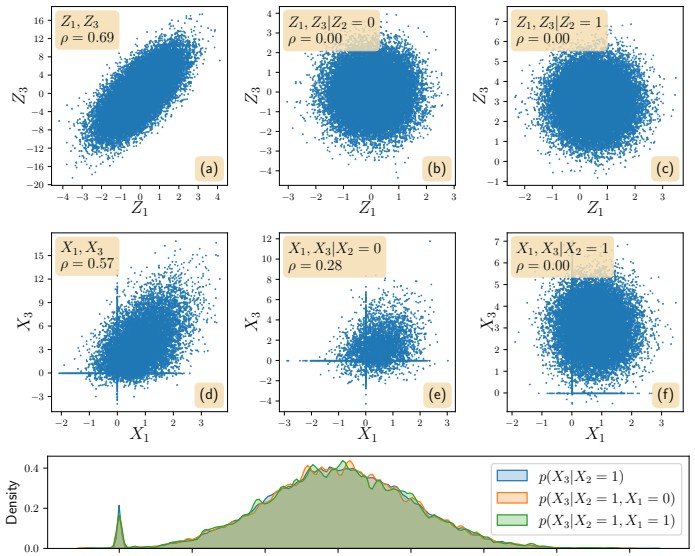

Figure 2: Above: scatterplots of $Z_1$; $Z_3$ and $X_1$; $X_3$ under different conditions. Below: density plot of $X_3$ under different conditions (vertical slices of scatters in (f)) to show $X_3 \perp\!\!\!\perp X_1 | X_2 = 1$. Kernel width is set to $= 0.02$ to prevent from oversmoothing.

axes (representing excessive zeros) and a distorted, non-elliptical distribution away from the axes, $X_1$ and $X_3$ are still dependent when $X_2 = 0$, because the zero entries of $X_2$ mix different values of $Z_2$. Nonetheless, this observation leads to an important insight: while a zero entry of $X_2$ may be noisy (i.e., may be technical), a non-zero entry of $X_2$ must be accurate, i.e., biological. Although the non-zero parts undergo a distribution distortion (which is why imputation fails), this distortion does not impact the estimation of conditional independence estimation, which only cares about each specific value. That is, conditioning on $X_2$ with any non-zero value $x_2 \neq 0$ is equivalent to conditioning on $Z_2$ with the same $x_2$ value, thereby eliminating the spurious dependence between $X_1$ and $X_3$. (f) shows the case where $X_2 = 1$. Note that even though there are still dropout points of $X_1$ and $X_3$ on axes in (f), these respective dropouts are also independent given $Z_2$. The identical density plots (vertical slices of (f)) in Figure 2 illustrates the conditional independence $X_1 \perp\!\!\!\perp X_3 | X_2 = 1$.

## 3    CAUSAL DISCOVERY IN THE PRESENCE OF DROPOUT

Building upon the intuition of "conditioning on non-zero entries" in Example 6, we now formally introduce our deletion-based CI test and the corresponding causal discovery methods.

**Assumptions 1.** We first list all the assumptions that may be used throughout this paper:

- *(A1.)* Common assumptions including causal sufficiency, Markov condition, and faithfulness over $\mathbf{Z} \cup \mathbf{X} \cup \mathbf{R}$, acyclicity of $G$, and a pointwise consistent CI testing method.
- *(A2.)* $\forall i, j \in [p]$, there is no edge $D_i \rightarrow Z_j$, i.e., dropout, as a technical artifact in the sequencing step, does not affect the genes' expressing process.
- *(A3.)* $\forall i \in [p]$, $D_i$ has at most one parent, $Z_i$, i.e., a gene's dropout can only be directly affected by the expression value of itself, if not completely at random.
- *(A4.)* For any variables $A, B \in \mathbf{X} \cup \mathbf{Z}, \mathbf{C} \subset \mathbf{X} \cup \mathbf{Z}, \mathbf{S} \subset [p]$, $A \not\!\perp\!\!\!\perp B | \mathbf{C}, \mathbf{R_S} \Rightarrow A \not\!\perp\!\!\!\perp B | \mathbf{C}, \mathbf{R_S} = \mathbf{0}$, i.e., dependencies conditioned on $\mathbf{R}$ are preserved in corresponding non-zero values.
- *(A5.)* For any conditioning set $\mathbf{S} \subset [p]$ involved in the required CI tests in the algorithm, $p(\mathbf{R_S} = \mathbf{0}) > 0$, i.e., asymptotically, the remaining sample size is large enough for test power.

We will elaborate more on these assumptions' plausibility in §4. Generally speaking, these assumptions are rather mild and are either theoretically testable or empirically supported by real data.

### 3.1    CONDITIONAL INDEPENDENCE ESTIMATION WITH ZEROS DELETED

Example 6 illustrates how the conditional (in)dependence relations may differ between true underlying expressions $\mathbf{Z}$ and dropout-contaminated $\mathbf{X}$. Now let us first investigate the scenario where the dropout issue is disregarded, and CI tests are directly conducted on the complete data points of $\mathbf{X}$:

**Proposition 1** (Bias to a denser graph). *Assume (A1), (A2). $\forall i \in [p], j \in [p], \mathbf{S} \subset [p]$, we have $Z_i \not\!\perp\!\!\!\perp Z_j | \mathbf{Z_S} \Rightarrow X_i \not\!\perp\!\!\!\perp X_j | \mathbf{X_S}$. The reverse direction does not hold in general, and holds only when $\mathbf{S} = \varnothing$.*

By Proposition 1, if one directly applies existing structure learning algorithms on observations $\mathbf{X}$ without dropout correction, the d-separation patterns entailed in the true GRN are generally undetectable. Consequently, the inferred graph tends to be much denser than the true one, posing a significant bias, given that gene regulatory relations are usually sparse (Dixit et al., 2016; Levine & Davidson, 2005).

To address this bias, we introduce the following clean and essential finding: ignoring the data points with zero values for the conditioned variables does not change the CI relations. While the remaining samples exhibit a different distribution from the true underlying distribution (as demonstrated in §2.4), they maintain identical conditional independence relations. Formally, we state the following theorem:

**Theorem 1** (Correct CI estimations). *Assume (A1), (A2), (A3), and (A4). For every $i \in [p], j \in [p], \mathbf{S} \subset [p]$, we have $Z_i \perp\!\!\!\perp Z_j | \mathbf{Z_S} \Leftrightarrow X_i \perp\!\!\!\perp X_j | \mathbf{Z_S}, \mathbf{R_S} = \mathbf{0}$.*

Note that in Theorem 1, the right hand side expresses the conditional independence relations as $X_i \perp\!\!\!\perp X_j | \mathbf{Z_S}, \mathbf{R_S} = \mathbf{0}$, rather than $Z_i \perp\!\!\!\perp Z_j | \mathbf{Z_S}, \mathbf{R_S} = \mathbf{0}$. This is because we only delete the samples with zeros for conditioned variables, while the zeros in estimands $X_i$ and $X_j$ are retained, and the underlying $Z_i$ and $Z_j$ remain unobservable. Generally speaking, for any $Z_i$ explicitly involved in a CI estimation, samples with the corresponding zeros must be deleted, i.e., $R_i = 0$ must be conditioned on. Alternatively, one may wonder what if the samples with zeros in all variables are deleted, not just the conditioning set. Interestingly, under assumption *(A3)*, both approaches are asymptotically equivalent, meaning that the CI relations can still be accurately recovered: $Z_i \perp\!\!\!\perp$

$Z_j|\mathbf{Z_S} \Leftrightarrow Z_i \perp\!\!\!\perp Z_j|\mathbf{Z_S}, \mathbf{R}_{i,j\cup\mathbf{S}} = \mathbf{0}$, though empirically, the latter may delete more samples and potentially reduce the statistical power of the test. It is also worth noting that in a more general setting where *(A3)* is dropped, the two approaches are not equivalent, and the latter provides a tighter bound for estimating conditional independencies. Further details will be discussed in Theorem 3.

## 3.2 CAUSAL DISCOVERY METHODS WITH DROPOUT CORRECTION

Now, we integrate the above consistent CI estimation into established causal discovery methods:

**Definition 1** (The general procedure for causal discovery with dropout correction). Perform any consistent causal discovery algorithm (e.g., PC (Spirtes et al., 2000)) based on CI relations estimated by test-wise deletion as in Theorem 1. Infer $Z_i \perp\!\!\!\perp_d Z_j|\mathbf{Z_S}$ if and only if $X_i \perp\!\!\!\perp X_j|\mathbf{Z_S}, \mathbf{R_S} = \mathbf{0}$, and use these d-separation patterns to infer the graph structure among $\mathbf{Z}$.

**Constraint-based methods.** The test-wise deletion can be seamlessly incorporated into any existing constraint-based methods for structure learning, as shown in Definition 1. Assumption *(A5)* ensures that asymptotically, for each CI relation to be tested, the remaining sample size after test-wise deletion remains infinite, and the consistency of the method is still guaranteed.

**Greedy score-based methods.** While the proposed GRNI on dropout data with integrated test-wise deletion in constraint-based methods is asymptotically consistent, empirical reliability may be limited due to *order-dependency* and *error propagation* of constraint-based methods (Colombo & Maathuis, 2014). In contrast, score-based methods generally search over the graph space and is immune to order-dependency, and thus may provide more empirically accurate results on large scale GRNs.

One typical score-based method is the Greedy Equivalence Search (GES (Chickering, 2002)) algorithm. GES uses a scoring function to assign a score to each directed acyclic graph (DAG) given data, and finds the optimal score by traversing over the space of CPDAGs. Consisting of a forward phase and a backward phase, GES iteratively performs edge additions and deletions in two phases respectively to optimize the total score until further improvement is not possible.

Score functions are typically assumed to exhibit three attributes: global consistency, local consistency, and decomposability. However, a closer examination on Chickering (2002)'s proof reveals that for GES's consistency, only a locally consistent score is needed. As another way of rendering faithfulness, local consistency enforces adding or deleting edges based on CI relations. This insight, as is also affirmed from the connection between BIC score and Fisher-Z test statistics (Lemma 5.1 of (Nandy et al., 2018)) and further echoed in (Shen et al., 2022), suggests that GES can be completely reframed as a constraint-based method using CI tests without a defined score. Thus, our deletion-based CI test can be easily incorporated into GES, ensuring asymptotic consistency and order-independence.

## 4 ON TESTABILITY OF THE CAUSAL DROPOUT MODEL

In §3, we propose a principled approach for GRNI on scRNA-seq data with dropouts, i.e., deleting samples with zeros for the conditioned variables in each CI test. The asymptotic correctness of this approach depends on assumptions listed in Assumptions 1. However, one may wonder whether these assumptions on the dropout mechanisms hold in practice. Therefore, in this section, we delve deeper into these assumptions and show that they can be either theoretically or empirically verified from data.

### 4.1 OVERVIEW OF THE ASSUMPTIONS

To enhance clarity, we first provide a detailed explanation of each assumption. The common assumptions for causal discovery in *(A1)* are thoroughly discussed in (Spirtes et al., 2000), and thus we focus more on the remaining four. Specifically, *(A2)* and *(A3)* are structural assumptions on the dropout mechanism, indicating that whether a gene is dropped out or not in the sequencing procedure (temporally later) does not affect the genes' expressions and interactions (temporally earlier), and that each gene's dropout can only be directly affected by the expression value of itself, not by other genes' dropouts or expression values. *(A4)* is called *faithful observability* in previous work on missing data (Strobl et al., 2018; Tu et al., 2019): while conditional independence means independence everywhere (on every conditioned value), conditional dependence only requires

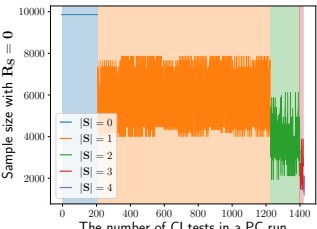

Figure 3: The remaining sample size after deleting zero conditioning samples, for each CI test during a run of PC on a real data (Dixit et al., 2016), with 15 genes and 9843 cells.

a dependence at some conditioned value. Thus *(A4)* is assuming that any dependence conditioned on $\mathbf{R}$ are preserved in corresponding non-zero values. This is reasonable as biological regulation is usually performed by genes' expression instead of non-expression, and it is unlikely that all the samples that capture the dependence happen to be dropped out. Lastly, *(A5)* assumes that for each required CI test during the algorithm, asymptotically there exists a sufficient number of samples for the test power. This assumption is empirically validated in real data, as shown in Figure 3, and is also validated by a synthetic experiment with varying dropout rates (Figure 12 in Appendix D.1).

## 4.2 VALIDATING AND DISCOVERING THE STRUCTURAL MECHANISM ON DROPOUT

Among the assumptions discussed above, the structural assumption *(A3)* concerning the dropout mechanism is of particular interest: each gene's dropout is solely affected by the gene itself and not by any other genes. While this assumption is in line with most of the existing parametric models (§2.3) and plausible in the context of scRNA-seq data, where mRNA molecules are reverse-transcribed independently across individual genes, it is still desirable to empirically validate this assumption based on the data. Therefore, here we propose a principled approach to validate this assumption. In other words, in addition to discovering the relations among genes, we also aim to discover the inherent mechanism to explain the dropout of each gene from the data.

Surprisingly, we have discovered that even without assumption *(A3)*, it is still possible to identify the regulations among $\mathbf{Z}$ and the dropout mechanisms for $\mathbf{R}$. The approach is remarkably simple: we perform causal structure search among $\mathbf{Z}$ using the procedure outlined in Definition 1, with only one modification: instead of deleting samples with zeros only for the conditioned variables, we delete samples with zeros for all variables involved in the CI tests. By doing so, the causal structure among $\mathbf{Z}$, representing the GRN, as well as the causal relationships from $\mathbf{Z}$ to $\mathbf{R}$, representing the dropout mechanisms, can be identified up to their identifiability upper bound. Formally, we have

**Definition 2** (Generalized GRN and dropout mechanisms discovery). Perform the procedure outlined in Definition 1, except for inferring $Z_i \perp\!\!\!\perp_d Z_j | \mathbf{Z_S}$ if and only if $Z_i \perp\!\!\!\perp Z_j | \mathbf{Z_S}, \mathbf{R}_{\mathbf{S} \cup \{i,j\}} = \mathbf{0}$.

**Theorem 2** (Identification of GRN and dropout mechanisms). *Assume (A1), (A2), (A4), and (A5). In the CPDAG among $\mathbf{Z}$ output by Definition 2, if $Z_i$ and $Z_j$ are non-adjacent, it implies that they are indeed non-adjacent in the underlying GRN, and for the respective dropout mechanisms, $Z_i$ does not cause $R_j$ and $Z_j$ does not cause $R_i$. On the other hand, if $Z_i$ and $Z_j$ are adjacent, then in the underlying GRN $Z_i$ and $Z_j$ are non-adjacent only in one particular case, and for dropout mechanisms, the existence of the causal relationships $Z_i \rightarrow R_j$ and $Z_j \rightarrow R_i$ must be naturally unidentifiable.*

Due to space limit, here we only give the above conclusion. For illustrative examples, proofs, and detailed elaboration on the identifiability in general cases without *(A3)*, please refer to Appendix C.

## 5 EXPERIMENTAL RESULTS

In this section, we conduct extensive experiments to validate our proposed method in §3, demonstrating that it is not only theoretically sound, but also leads to superior performance in practice. We provide the experimental results on linear SEM simulated data, more 'realistic' synthetic and curated scRNA-seq data, and real-world experimental data in §5.1, §5.2, and §5.3, respectively[1].

### 5.1 LINEAR SEM SIMULATED DATA

**Methods and simulation setup.** We assess our method (testwise deletion) by comparing it to other dropout-handling approaches on simulated data, including MAGIC Van Dijk et al. (2018) for imputation, mixedCCA Yoon et al. (2020) as parametric model, and direct application of algorithms on full samples (without deleting or processing dropouts, corresponding to Proposition 1). We also report the results on the true underlying $\mathbf{Z}$, denoted as Oracle*, to examine the performance gap between these methods and the best case (but rather impossible) without dropouts. PC Spirtes et al. (2000) and GES (Chickering, 2002) with FisherZ test are used as the base causal discovery algorithms. We randomly generate ground truth causal structures with $p \in \{10, 20, 30\}$ nodes and degree of 1 to 6. Accordingly, $\mathbf{Z}$ is simulated following random linear SEMs in two cases: jointly Gaussian, and Lognormal distributions to better model count data (Bengtsson et al., 2005). We apply three different types of dropout mechanisms on $\mathbf{Z}$ to obtain $\mathbf{X}$: (1) dropout with the fixed rates, (2) truncating low expressions to zero, and (3) dropout probabilistically determined by expression, which are described in Examples 1 to 3, respectively. We report the structural Hamming distance (SHD) wrt. the true and estimated CPDAGs calculated from 5 random simulations. More details are available in Appendix D.1.

---

[1]Codes are available at `https://github.com/MarkDana/scRNA-Causal-Dropout`.

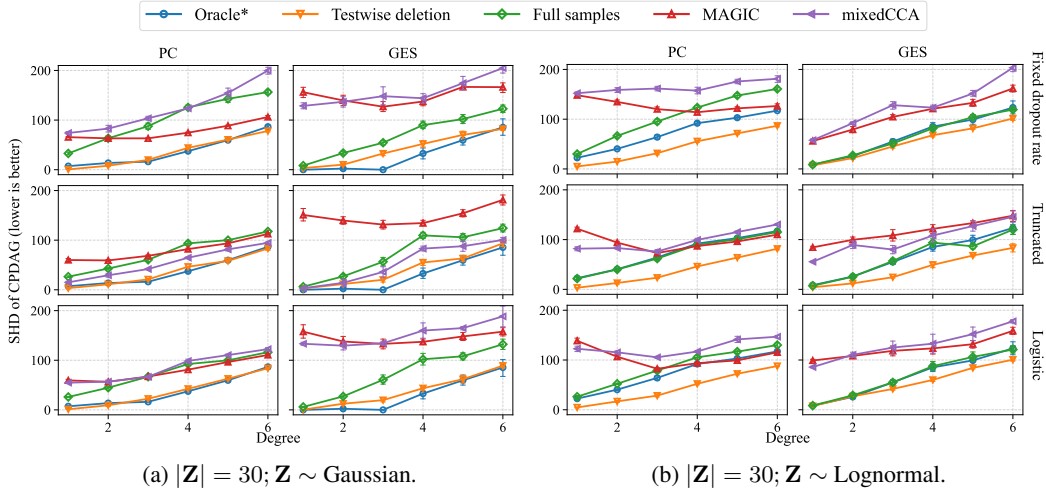

Figure 4: Experimental results (SHDs of CPDAGs) of 30 variables on simulated data, where three dropout mechanisms are considered. The variables $\mathbf{Z}$ follow Gaussian or Lognormal distribution.

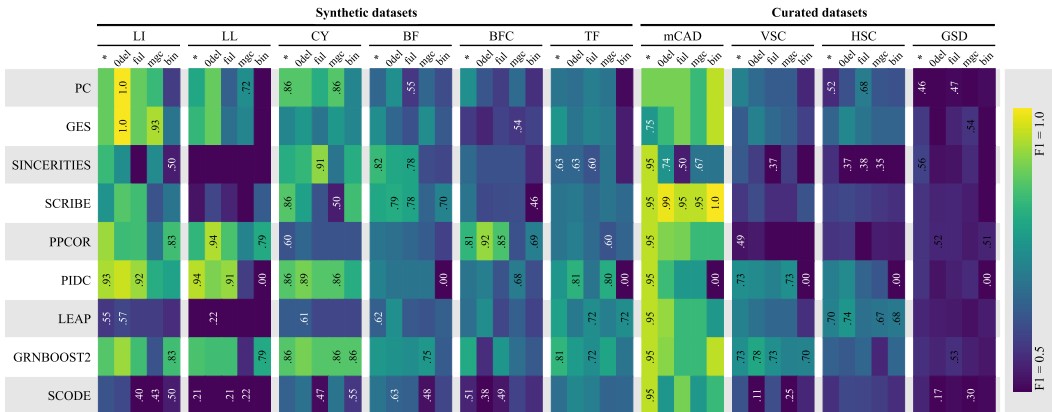

Figure 5: Experimental results (F1-scores of estimated skeleton edges) on BoolODE simulated data and BEELINE benchmark framework (Pratapa et al., 2020). The 9 rows correspond to PC, GES, and 7 other GRNI-specific SOTA algorithms benchmarked. The 10 colored column blocks correspond to all 6 synthetic and 4 curated datasets in (Pratapa et al., 2020). The 5 column strips in each block correspond to different dropout-handling strategies. Cell colors indicate the corresponding values (brighter is higher, i.e., better). The maximum and minimum of each strip are annotated.

**Experimental results.** The results of 30 nodes are provided in Figure 4. Our method (1) consistently outperforms other competitors across most settings, particularly when using PC as the base algorithm, and (2) leads to performance close to the oracle. Directly applying algorithms on full samples often performs poorly due to potential false discoveries as suggested by Proposition 1. Notably, mixedCCA performs relatively well in the Gaussian and truncated dropout setting (as a parametric model with no model misspecification) but is still outperformed by our method. This further validates our method's effectiveness and flexibility. For more experimental results, including results in different scales, on different metrics, and with varying dropout rates, please kindly refer to Appendix D.1.

## 5.2 REALISTIC BOOLODE SYNTHETIC AND CURATED DATA

While synthetic data is preferred for assessment thanks to ground-truth graphs, to truly gauge real-world applicability, we aim to conduct simulations that resemble scRNA-seq data more closely than just linear SEM. Furthermore, we seek to investigate the efficacy of our zero-deletion approach when integrated into algorithms specifically designed for scRNA-seq data, rather than just general causal discovery methods like PC and GES. To accomplish this, we conduct an array of experiments based

on the widely recognized BEELINE framework (Pratapa et al., 2020), which offers a scRNA-seq data simulator called BoolODE, and benchmarks various SOTA GRNI-specific algorithms.

Following the BEELINE paradigm, for data simulation, we use the BoolODE simulator that basically simulates gene expressions with pseudotime indices from Boolean regulatory models. We simulate all the 6 synthetic and 4 literature-curated datasets in (Pratapa et al., 2020), each with 5,000 cells and a 50% dropout rate, following all the default hyper-parameters. For algorithms, in addition to PC and GES, we examined all SOTA algorithms (if executable; with default hyper-parameters) benchmarked in (Pratapa et al., 2020). For each algorithm, five different dropout-handling strategies are assessed, namely, oracle*, testwise deletion, full samples, imputed, and binarization (Qiu, 2020). Method details, e.g., how is zero deletion incorporated into various algorithms, can be found in Appendix D.2.

The F1-scores of the estimated skeleton edges are shown in Figure 5. We observe that: 1) Dropouts do harm to GRNI. Among the 90 dataset-algorithm pairs (9 algorithms × 10 datasets), on 65 of them (72%), 'full samples' (with dropouts) performs worse than 'oracle'. 2) Existing dropout-handling strategies (imputation and binarization) don't work well. Among the 90 pairs, 'imputed' and 'binary' is even worse than 'full samples' on 45 (50%) and 58 (64%) of them, respectively, i.e., they may even be counterproductive. As discussed in §2.4, such strategies may indeed introduce additional bias. 3) The proposed zero-deletion is effective in dealing with dropouts, with consistent benefits across different integrated algorithms and on different datasets. Among the 90 pairs, 'zero-deletion' is better than 'full samples' and than 'imputed' on 64 (71%) and 61 (68%) of them, respectively, i.e., it effectively helps GRNI with dropouts, and is more effective than imputation in dropout-handling.

### 5.3 REAL-WORLD EXPERIMENTAL DATA

To examine the efficacy of our approach in a real-world setting, we applied our method on gene expression data collected via single-cell Perturb-seq (Dixit et al., 2016). As in prior work Saeed et al. (2020); Dixit et al. (2016), we focus on the bone-marrow dendritic cells (BMDC) in the unperturbed setting (9843 cells in total), and on 21 transcription factors believed to be driving differentiation in this tissue type Dixit et al. (2016); Garber et al. (2012). We compared the predicted causal edges with known relations derived as a union of statistically significant predictions from interventional data Dixit et al. (2016) and prior knowledge Han et al. (2018). For this analysis, we ran a baseline of the PC algorithm with all the samples (PC-full), compared to the PC algorithm with testwise deletion (PC-test-del). From the figure, one can appreciate that PC-full introduces numerous edges that are not backed by prior knowledge (labeled red in Figure 6). For example, the E2f1 → Cebpbp causal link is opposite of what has been reported in the literature

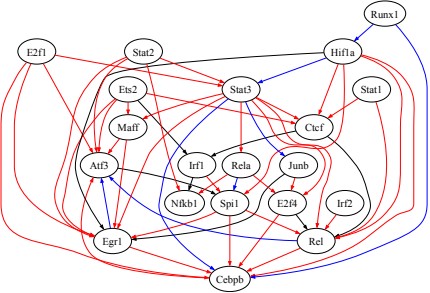

Figure 6: Experimental results on 21 key genes from Pertub-seq data (Dixit et al., 2016). Red edges are returned by PC-full but not by PC-test-del. Blue and black edges are returned by PC-test-del. Specifically, blue edges are the regulatory interactions priorly known (Dixit et al., 2016), while black edges are not priorly known.

Gutsch et al. (2011). Furthermore, Stat3 and E2f1 are known to cooperate in downstream regulation, rather than E2f1 being a cause of Stat3 Hutchins et al. (2013). On the contrary, majority of the edges that testwise deletion provides are previously known (labeled in blue), with a few that are not in the list of previously seen interactions that we used (labeled in black). For experimental details and results on another two real-world experimental datasets, please kindly refer to Appendix D.3.

## 6 CONCLUSION AND DISCUSSIONS

Building upon common understanding of dropout mechanisms, we develop the causal dropout model to characterize these mechanisms. Despite the non-ignorable observed zeros resulting from dropouts, we develop a testwise-deletion procedure to reliably perform CI test, which can be seamlessly integrated into existing causal discovery methods to handle dropouts and is asymptotically correct under mild assumptions. Furthermore, our causal dropout model serves as a systematic framework to verify if the qualitative mechanism of dropout studied in the literature is valid, and learn such a mechanism from observations. Extensive experiments on simulated and real-world datasets demonstrate that our method leads to improved performance in practice. A possible limitation is the decreasing sample size after testwise deletion, and future work includes developing a practical method to resolve it.

ACKNOWLEDGEMENTS

This material is based upon work supported by the AI Research Institutes Program funded by the National Science Foundation under AI Institute for Societal Decision Making (AI-SDM), Award No. 2229881. The project is also partially supported by the National Institutes of Health (NIH) under Contract R01HL159805, and grants from Apple Inc., KDDI Research Inc., Quris AI, and Infinite Brain Technology. Petar Stojanov was supported in part by the National Cancer Institute (NCI) grant number: K99CA277583-01, and funding from the Eric and Wendy Schmidt Center at the Broad Institute of MIT and Harvard.

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

# Appendix

# Table of Contents

# A  PROOFS OF MAIN RESULTS

## A.1  PROOF OF PROPOSITION 1

**Proposition 1** (Bias to a denser graph). *Assume (A1), (A2). $\forall i \in [p], j \in [p], \mathbf{S} \subset [p]$, we have $Z_i \not\perp Z_j | \mathbf{Z_S} \Rightarrow X_i \not\perp X_j | \mathbf{X_S}$. The reverse direction does not hold in general, and holds only when $\mathbf{S} = \varnothing$.*

*Proof of Proposition 1.* The $\Rightarrow$ direction: By Markov condition in *(A1)*, we have $Z_i \not\perp Z_j | \mathbf{Z_S} \Rightarrow Z_i \not\perp_d Z_j | \mathbf{Z_S}$. Consider the open path $Z_i - \cdots - Z_j$, by *(A2)* all non-colliders are not in $\mathbf{X_S}$ and all colliders have descendants in $\mathbf{X_S}$, and thus $X_i \leftarrow Z_i - \cdots - Z_j \rightarrow X_j$ is still open, i.e., $X_i \not\perp_d X_j | \mathbf{X_S}$. By faithfulness in *(A1)*, $X_i \not\perp_d X_j | \mathbf{X_S} \Rightarrow X_i \not\perp X_j | \mathbf{X_S}$.

The $\Leftarrow$ direction doesn't hold: Because the non-colliders in $\mathbf{Z_S}$ cannot be conditioned in children $\mathbf{X_S}$, $X_i \perp X_j | \mathbf{X_S}$ only when every path connecting $Z_i$ and $Z_j$ has a collider, i.e., $Z_i \perp Z_j$. $\qquad\square$

## A.2  PROOF OF THEOREM 1

**Theorem 1** (Correct CI estimations). *Assume (A1), (A2), (A3), and (A4). For every $i \in [p], j \in [p], \mathbf{S} \subset [p]$, we have $Z_i \perp Z_j | \mathbf{Z_S} \Leftrightarrow X_i \perp X_j | \mathbf{Z_S}, \mathbf{R_S} = \mathbf{0}$.*

*Proof of Theorem 1.* The $\Rightarrow$ direction: By faithfulness in *(A1)*, $Z_i \perp Z_j | \mathbf{Z_S} \Rightarrow Z_i \perp_d Z_j | \mathbf{Z_S}$. By *(A3)*, the blocked paths remain blocked for the two children, i.e., $X_i \perp_d X_j | \mathbf{Z_S}$. Since conditioning on $\mathbf{R_S}$ does not introduce any collider, we further have $X_i \perp_d X_j | \mathbf{Z_S}, \mathbf{R_S}$. By Markov condition in *(A1)*, $X_i \perp_d X_j | \mathbf{Z_S}, \mathbf{R_S} \Rightarrow X_i \perp X_j | \mathbf{Z_S}, \mathbf{R_S}$, which implies $X_i \perp X_j | \mathbf{Z_S}, \mathbf{R_S} = \mathbf{0}$.

The $\Leftarrow$ direction: Show by its contrapositive. By Markov condition in *(A1)*, $Z_i \not\perp Z_j | \mathbf{Z_S} \Rightarrow Z_i \not\perp_d Z_j | \mathbf{Z_S}$. By *(A2)*, $\mathbf{R_S}$ does not contain any non-collider on the open path connecting $Z_i$ and $Z_j$, and thus still $Z_i \not\perp_d Z_j | \mathbf{Z_S}, \mathbf{R_S}$. By *(A3)*, the open paths remain for the two children, i.e., $X_i \not\perp_d X_j | \mathbf{Z_S}, \mathbf{R_S}$. By faithfulness in *(A1)*, $X_i \not\perp X_j | \mathbf{Z_S}, \mathbf{R_S}$. By *(A4)*, $X_i \not\perp X_j | \mathbf{Z_S}, \mathbf{R_S} = \mathbf{0}$. $\quad\square$

A.3 PROOFS OF RESULTS IN §4.2

For results in §4.2 (including Proposition 2 and Theorems 2 and 3) regarding the discovery and validation of the dropout mechanisms within a general causal dropout model without assumption *(A3)*, please refer to Appendix C where we will first provide the necessary background and introduce the general framework to support our findings.

# B DISCUSSIONS

## B.1 EXISTING PARAMETRIC MODELS AS INSTANCES OF THE CAUSAL DROPOUT MODEL

In §2.3, we discussed several existing parametric models as specific instances of our proposed causal dropout model. Here we give some detailed analysis on them:

1. Dropout with the fixed rates (Example 1). $D_i \sim \text{Bernoulli}(p_i)$, i.e., the gene $Z_i$ gets dropped out with a fixed probability $p_i$ across all individual cells. The representative models in this category are the zero-inflated models (Pierson & Yau, 2015; Kharchenko et al., 2014; Saeed et al., 2020; Yu et al., 2023; Min & Agresti, 2005) (or with a slight difference, the hurdle models (Finak et al., 2015; Qiao et al., 2023)), where sequenced data $X_i$ are assumed to follow a mixture distribution with two components: one point mass at zero for dropouts, and one common distribution, e.g., Gaussian, Poisson or negative binomial, for gene counts. Biologically, dropouts with the fixed rates can be explained by the random sampling of transcripts during library preparation – regardless of the true expressions of genes. Graphically, in this case the edge $Z_i \rightarrow D_i$ is absent. It is worth noting that in some (e.g., the Michaelis-Menten (Andrews & Hemberg, 2019)) models, the dropout rate is determined by the average expression of the gene, i.e., $p_i = f(\mathbb{E}[Z_i])$. In this case however, the edge $Z_i \rightarrow D_i$ is still absent, as apparently $Z_i \perp\!\!\!\perp D_i$ (this $p_i$ is fixed across all cells).
2. Truncating low expressions to zero (Example 2). $D_i = \mathbb{1}(Z_i < c_i)$, i.e., the gene $Z_i$ gets dropped out in cells whenever its expression is lower than a threshold $c_i$. A typical kind of such truncation models is, for simple statistical properties, the truncated Gaussian copula (Fan et al., 2017; Yoon et al., 2020; Chung et al., 2022), where $\mathbf{Z}$ are assumed to be joint Gaussian (usually with additional assumptions e.g., standardized), and the covariance among $\mathbf{Z}$ is estimated from $\mathbf{X}$. Biologically, such truncation thresholds $c_i$ (quantile masking (Jiang et al., 2022)) can be explained by limited sequencing depths. Graphically, in this case the edge $Z_i \rightarrow D_i$ exists.
3. Dropout probabilistically determined by expressions (Example 3). $D_i \sim \text{Bernoulli}(F_i(\beta_i Z_i + \alpha_i))$, where usually $F_i$ are strictly monotonically increasing functions in range $[0, 1]$, and $\beta_i, \alpha_i \in \mathbb{R}$ are parameters with $\beta_i < 0$, i.e., a gene $Z_i$ may be detected (or not) in every cell, while the higher it is expressed in a cell, the less likely it will get dropped out. $F_i$ is typically chosen as CDF of some common distributions, e.g., probit or logistic (Cragg, 1971; Liu, 2004; Miao et al., 2016). Biologically, the mechanism can be explained by inefficient amplification. Graphically, the edge $Z_i \rightarrow D_i$ also exists, and is non-deterministic.

Specifically regarding the several existing zero-inflated models (Pierson & Yau, 2015; Kharchenko et al., 2014; Saeed et al., 2020; Yu et al., 2023; Min & Agresti, 2005) (or related hurdle models (Finak et al., 2015; Qiao et al., 2023)) that can be seen as parametric instances within our proposed causal dropout model, they align with Example 1, where each gene $Z_i$ experiences dropout with a fixed probability $p_i$ across all cells. However, it is worth noting that there are also some other zero-inflated models and variations that fall outside the scope of our causal dropout model. We categorize these deviations based on the following three reasons:

1. Dropout is influenced by other covariates, rather than solely by the gene itself. In certain models, the excessive zero rate is not fixed across all cells, meaning that $Z_i \not\perp\!\!\!\perp D_i$. Furthermore, this dependency cannot be explained solely by $Z_i$, as $\mathbf{Z} \backslash Z_i \not\perp\!\!\!\perp D_i | Z_i$. An example of such a model is one that models the zero rate using a logit link, as in (Workie & Azene, 2021): $D_i \sim \text{Bernoulli}(p_i)$ with $p_i = \beta_i^\mathsf{T} \mathbf{Z}$, where $\beta_i$ is the parameter vector in $\mathbb{R}^p$. Here, the non-zero entries of $\beta_i$ are not limited to the $i$-th entry, and all other entries are also considered as "parents" of $Z_i$ in GRN (Choi et al., 2020). This aligns with our findings in §4.2 and Appendix C.2, where the causal effects within the GRN or as dropout mechanisms are unidentifiable.

2. Measurement errors are involved. In our model, $X_i = (1 - D_i) * Z_i$, indicating that the latent variable $Z_i$ is partially observed, with the non-zero observations being the true values. However, in representative models such as the post Poisson model (Xiao et al., 2022; Saeed et al., 2020), the data generating process is formulated as follows:

$$X_i \sim \begin{cases} \text{Poisson}(Z_i) & \text{w.p. } p_i \\ 0 & \text{w.p. } 1 - p_i \end{cases} \tag{B.1}$$

In this case, even if $p_i$ is fixed across all cells, it does not fit into our model as the non-zero parts of the observation $X_i$ do not reflect the true values of $Z_i$ due to the post-Poisson noise. This scenario should be addressed separately as the problem of causal discovery in the presence of *measurement error* (Fuller, 2009; Pearl, 2012; Kuroki & Pearl, 2014; Scheines & Ramsey, 2016; Dai et al., 2022).

3. Excessive zeros are incorporated into the latent generation process, rather than being modeled as a post-sequencing procedure. In our model, we have two sets of variables: $\mathbf{Z}$, corresponding to the true underlying expressions, and $\mathbf{X}$, corresponding to the observations. E.g., in (Xiao et al., 2022; Saeed et al., 2020), each $X_i$ conditioned on $Z_i$ is assumed to follow a zero-inflated Poisson distribution. However, in certain models, there is no such differentiation. The observations $\mathbf{X}$ are assumed to be exactly the true expressions, meaning there are no technical dropouts. For example, in (Choi et al., 2020), the true expression of each gene, conditioned on its parents in the GRN, directly follows a zero-inflated Poisson distribution, i.e., these zeros are inflated in the underlying genes interactions, not in a post sequencing procedure. It is important to note that this type of model is not designed to address the dropout issue but rather to provide a more accurate characterization (beyond just Poisson) of the excessive zeros in the true gene expressions.

### B.2 Relation to Imputation Methods and the Missing Data Problem

In §2.4, we examined the potential theoretical issues associated with imputation methods and demonstrated the general unidentifiability of the true underlying joint distribution $p(\mathbf{Z})$ from the observational distribution $p(\mathbf{X})$. Now, we delve deeper into the derivation of this result and provide a more comprehensive discussion of the broader missing data problem.

Firstly, according to the missing data literature, the underlying joint distribution $p(\mathbf{Z})$ is *irrecoverable* due to the *self-masking* dropout mechanism (Enders, 2022; Little & Rubin, 2019; Mohan et al., 2013; Shpitser, 2016; Mohan, 2018). To "fill in the holes" on any subset of variables $\mathbf{S}$ from non-zero observations, i.e., to recover the joint distribution $p(\mathbf{Z_S})$ from $p(\mathbf{X_S}|\mathbf{R_S} = \mathbf{0})$ (which is $p(\mathbf{Z_S}|\mathbf{R_S} = \mathbf{0})$), it generally requires $p(\mathbf{Z_S})$ to be factorizable into components where the missingness is *ignorable*, i.e., $p(\cdot) = p(\cdot|\mathbf{R}_\cdot = \mathbf{0})$, unbiasedly estimable from non-zero parts. By induction, at least one gene's marginal distribution $p(Z_i)$ should be recoverable, which is however impossible: since $Z_i$ directly affects the dropout by itself, $Z_i$ and $D_i, R_i$ are dependent conditioning on any other variables, owing to the presence of the directed edges between $Z_i$ and $D_i, R_i$ in Figure 1. This irrecoverability persists even if the dropout mechanism is assumed to be parametrically fixed, as shown by the counterexample Example 4.

Secondly, what distinguishes dropout data from missing data is that the missing entries cannot be precisely located, i.e., the

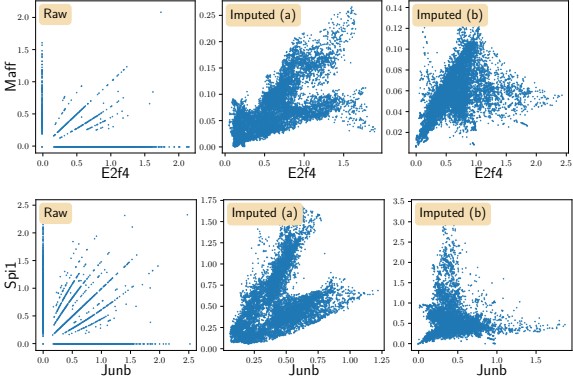

Figure 7: Examples illustrating the introduction of spurious relationships through imputation. The two rows correspond to two gene pairs, namely (E2f4, Maff) and (Junb, Spi1), obtained from unperturbed cells in the data by (Dixit et al., 2016). The three columns represent the raw counts after normalization and the imputed data using MAGIC (Van Dijk et al., 2018) with (b) incorporation of all genes or (c) considering only the 24 important genes as discussed in §5.3. It can be observed that imputation introduces spurious relationships, such as the presence of fork-shaped and highly nonlinear associations. Furthermore, the disparity between (b) and (c) demonstrates that the selection of genes significantly impacts the outcomes of imputation.

**D** indicators are latent. In scRNA-seq data, we only observe zero entries but cannot differentiate between biologically unexpressed genes and technical zeros. Most existing imputation methods treat all zeros as "missing holes" resulting from technical dropout and fill them in using information from non-zero entries. However, given that biologically unexpressed genes are ubiquitous in cells, such imputation can introduce false signals and spurious relationships (Andrews & Hemberg, 2018), as illustrated in Figure 7. Notably, many imputation methods only evaluate their accuracy under the fixed-dropout-rate scheme (Jiang et al., 2022) as in Example 1, where dropout happens *completely-at-random* (CAR) (Little & Rubin, 2019), i.e., $\mathbf{Z} \perp\!\!\!\perp \mathbf{D}$. However, even in this relatively simpler case (without the self-masking as in the first reason), $p(\mathbf{Z})$ remains irrecoverable, as shown by the counterexample Example 4.

And thirdly, even if one is willing to make restrictive assumptions to render $p(\mathbf{Z})$ theoretically recoverable, imputation methods maynot be the most suitable choice for gene regulatory network inference, which is inherently a structure learning task. This is because imputation has a different inductive bias compared to structure learning: it focuses primarily on the unconditional (pairwise) dependencies and is better suited for tasks such as differential expression analysis. However, for structure learning, conditional dependencies are more important in order to capture the underlying gene regulatory relationships (Saeed et al., 2020; Gao et al., 2022). As interestingly shown by (Gao et al., 2022), in the linear Gaussian setting where covariance matrix is a sufficient statistic, imputation methods and a method specifically for causal discovery are compared on recovering the structure with missing data. Even if imputation methods estimate a more accurate covariance matrix (in terms of a smaller Frobenius norm), by inputting the estimated covariance matrices into causal discovery method e.g., PC, their performance on structure learning (in terms of SHD to the true graph) is worse than the other method specifically designed for structure learning. Therefore, imputation methods may not provide the optimal inductive bias required for accurate gene regulatory network inference.

In light of the demonstrated unidentifiability of $p(\mathbf{Z})$ using the decomposition technique from the missing data literature, we now delve into the missing data problem and its relevance to our work. Missing data entries are frequently encountered in real-world data, such as unanswered questions in a questionnaire. These missingness patterns can be categorized into three main types: Missing Completely At Random (MCAR), Missing At Random (MAR), and Missing Not At Random (MNAR). Data are considered MCAR when the cause of missingness is purely random, such as instances where the missing rate is fixed across all samples (as in Example 1) and entries are deleted due to random computer errors. On the other hand, data are deemed MAR when the missingness is independent with the corresponding underlying variables given all other fully observed variables (i.e., no missing entries at all), meaning that the missingness can be completely explained by the fully observed variables. For example, consider the gender wage gap study that measures two variables: gender and income, where gender is always observed and income has missing entries. In this scenario, MAR missingness would occur if men are more reluctant than women to disclose their income. Given the fully observed gender, the missing entries are independent with the income. Lastly, data that do not fall under either MCAR or MAR are classified as MNAR.

To handle missing data, a trivial approach is list-wise deletion, where all samples with missing entries are discarded. Another common strategy is data imputation, often performed through expectation maximization (EM) techniques. In recent years, there has been a growing interest in understanding missing data from a causal perspective, with contributions from various studies (Enders, 2022; Little & Rubin, 2019; Mohan et al., 2013; Shpitser, 2016). By modeling the missingness mechanisms within a causal graphical model, researchers have explored the conditions for recoverability of the underlying distribution and developed techniques for recovery. It has been shown that list-wise deletion can only recover the true distribution when the missing mechanism is MCAR. EM-based imputation methods can unbiasedly recover the true distribution only under the MAR missingness. For MNAR, some cases can be addressed through techniques like factorization or probability reweighting (Mohan et al., 2013), but certain situations are theoretically irrecoverable, such as the well-known *self-masking* case (Mohan, 2018). Some other studies propose that even if the true underlying distribution is irrecoverable, the conditional independence (CI) relations can still be identifiable through structure learning. This is achieved by employing test-wise deletion, where incomplete records of variables involved in each CI test are removed (Strobl et al., 2018; Tu et al., 2019).

In our specific task, we do not encounter missing entries explicitly marked as, for example, `nan` in the data matrix. Instead, we only observe zeros, which can be attributed to either technical or biological reasons that are unknown to us. If we treat all zeros as missing entries, self-masking edges exist

for each variable. Intriguingly, even in this challenging scenario, according to the current biological understanding to the dropout mechanism, our proposed causal dropout model can systematically address the dropout issue based on the current biological understanding of the dropout mechanism. As a result, the true graph structure can be consistently estimated. Lastly, it is worth noting that within our proposed framework, as illustrated in Definition 1, test-wise deletion only removes samples with zeros for the conditioned variables, rather than all variables. This approach deviates from the conventional notion of test-wise deletion, and we will further discuss this distinction in Appendix C.2.

## C ON A GENERAL CAUSAL DROPOUT MODEL WITH RELAXED ASSUMPTIONS

Among the assumptions delineated in §4.1, the structural assumption *(A3)* concerning the dropout mechanism is of particular interest: each gene's dropout is solely affected by the gene itself and not by any other genes. While this assumption is in line with most of the existing parametric models (§2.3) and plausible in the context of scRNA-seq data, where mRNA molecules are reverse-transcribed independently across individual genes, it is still desirable to empirically validate this assumption based on the data. In other words, except for discovering the relations among genes, we also aim to discover the inherent mechanism to explain the dropout of each gene from the data. Surprisingly, as in §4.2, we propose a principled and simply approach to do so: the causal structure among $\mathbf{Z}$, representing the GRN, as well as the causal relationships from $\mathbf{Z}$ to $\mathbf{R}$, representing the dropout mechanisms, can be identified up to their identifiability upper bound. Now here we will first give a detailed motivation and elaboration in Appendices C.1 and C.2, and then give the proofs in Appendix C.3.

### C.1 ELABORATION ON THEOREM 3 WITH AN ILLUSTRATIVE EXAMPLE

Theorem 2 gives a principled approach to identify both the GRN and the dropout mechanisms up to their identifiability upper bound even without assumption Assumptions 1. Before we move into the proof in Appendix C.2, here we first explain what the identifiability upper bound is, and why.

To check whether gene $Z_i$ affects dropouts of another gene $Z_j$, it basically requires us to view $R_j$ also as random variables, and to test for conditional independence and see whether there exists any variables set that can d-separate $Z_i$ from $R_j$. However, there exists a natural identifiability upper bound:

**Proposition 2** (Identifiability upper bound of dropout mechanisms). *If $Z_i$ and $Z_j$ are adjacent in GRN, then whether they affect each other's dropout, i.e., whether edges $Z_i \to R_j$ and $Z_j \to R_i$ exist, is unidentifiable.*

This is because to condition on $Z_j$, $R_j$ must also be conditioned but it is already in the estimands. Due to this, we only focus on identifying the non-adjacent pairs of variables in GRN, i.e., to recover the conditional independencies among $\mathbf{Z}$. We have the following results:

**Theorem 3** (Observed independencies are correct independencies). *Assume (A1), (A2), (A4). $\forall i, j, \mathbf{S}$,*

$$X_i \perp\!\!\!\perp X_j | \mathbf{X_S} \overset{①}{\Rightarrow} X_i \perp\!\!\!\perp X_j | \mathbf{Z_S}, \mathbf{R_S} = \mathbf{0} \overset{②}{\Rightarrow} Z_i \perp\!\!\!\perp Z_j | \mathbf{Z_S}, \mathbf{R_{S \cup \{i,j\}}} = \mathbf{0} \overset{③}{\Rightarrow} Z_i \perp\!\!\!\perp Z_j | \mathbf{Z_S}. \quad \text{(C.1)}$$

Theorem 3 provides insights that align with the previous Proposition 1: although the conditional independencies inferred from observational data with dropouts are "rare", they are accurate. The first three terms are corresponding to Proposition 1, Definition 1, and Definition 2 respectively, and the fourth is oracle. When the structural assumption *(A3)* is in the play, the converse directions of ② and ③ also hold, as stated in Theorem 1. However, in the general setting without *(A3)*, these implications hold only in one direction. Progressing from left to right, the testing approaches provide increasingly tighter bounds on the correct underlying conditional independence relations,[2] while also in empirical terms, they are less data-efficient due to the increased deletion of samples.

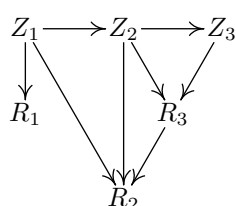

Figure 8: An illustrative example for dropout mechanism identification.

Consider when we have identified a non-adjacent pair $Z_i$ and $Z_j$, and the question arises: How can we determine the presence of causal relationships $Z_i \to R_j$ and $Z_j \to R_i$? Should we conduct a conditional independence (CI) test between $Z_i$ and $R_j$ as suggested in Proposition 2? However, engaging in such a procedure would lead us astray. Let

---

[2]As for how tight it theoretically can be, see Appendix C.2.

us examine Figure 8 as an example. We have identified $Z_1$ and $Z_3$ as non-adjacent based on the conditional independence $Z_1 \perp\!\!\!\perp Z_3 | Z_2, \mathbf{R}_{1,2,3} = \mathbf{0}$. Then we test for causes of $R_3$. However, even if $Z_1 \to R_3$ does not exist, we cannot establish their independence: $Z_1 \not\perp\!\!\!\perp R_3 | Z_2, \mathbf{R}_{1,2} = \mathbf{0}$ due to the presence of the collider $R_2$. So, what should be our course of action? Surprisingly, the answer already lies before us: $Z_1 \to R_3$ must not exist; otherwise, the independence $Z_1 \perp\!\!\!\perp Z_3 | Z_2, \mathbf{R}_{1,2,3} = \mathbf{0}$ would not have been identified initially, owing to the presence of the collider $R_3$. In other words, the non-adjacency of $Z_1 \to R_3$ is not established through a CI test, but rather through logical deduction.

Finally, we have come to the result in Theorem 2. The sparse causal graphs identified from real-world data also suggests the sparse dropout mechanisms, which, from the empirical view, supports our *(A3)*.

## C.2 MOTIVATION ON THE GENERAL CAUSAL DROPOUT MODEL

Generally speaking, we discover the causes of zero indicators $\mathbf{R} = \{R_i\}_{i=1}^p$ by also viewing them as random variables and conducting causal structure search among them. As is mentioned in §2.2, the causal model in Figure 1 contains redundant deterministic edges and auxiliary variables just for convenience of notations. Now given the already built basic understanding, we proceed with an equivalent but more compact model, with only two parts of variables, $\mathbf{Z}$ and $\mathbf{R}$.

Given that $R_i = \mathbb{1}(X_i = 0) = D_i$ OR $\mathbb{1}(Z_i = 0)$, it follows that the edge $Z_i \to R_i$ always exists. Notably, even in the presence of dropout that happens completely at random, i.e., $Z_i \perp\!\!\!\perp D_i$ as shown in Example 1, the edge $Z_i \to R_i$ remains, due to the $\mathbb{1}(Z_i = 0)$ term. In this section, we drop *(A3)* and seek to empirically verify it, while retaining the assumptions *(A1)*, *(A2)*, *(A4)*, and *(A5)* (specifically, we continue to assume *(A2)*, i.e., dropouts do not affect genes underlying expressions).

To conduct causal structure search on variables $\mathbf{Z} \cup \mathbf{R}$, we need to identify d-separation patterns by performing CI tests. However, due to the definition of this causal model, some d-separation patterns naturally cannot be read off from the data. Specifically, the following rules should be followed:

**Proposition 3** (Testability of CI queries). *A CI query in form $A \perp\!\!\!\perp ? B | \mathbf{C}$ is testable if and only if:*

1. *$\{R_i : i \in [p] \text{ s.t. } Z_i \in \{A, B\} \cup \mathbf{C}\} \subset \mathbf{C}$, and*
2. *$A \neq B$ and $A, B \notin \mathbf{C}$,*

*where 1. means that any underlying $\mathbf{Z}$ variables involved in the test is testable (accessible) only when the corresponding $\mathbf{R}$ variable is also conditioned, i.e., keep only the non-zero samples. 2., together with 1., rules out the queries e.g., $R_i \perp\!\!\!\perp ? R_j | Z_j$ or $Z_j \perp\!\!\!\perp ? R_j$ (i.e., $R_j$ cannot be further conditioned).*

Proposition 3 gives a sufficient and necessary condition for a CI query to be testable from observational data $\mathbf{X}$. Now we further give the criteria to asymptotically estimate these CI queries from data $\mathbf{X}$. For any CI query that is valid according to Proposition 3, it can be estimated by:

**Proposition 4** (Testing CI queries from data). *For each variable involved in a CI query $A \perp\!\!\!\perp ? B | \mathbf{C}$:*

1. *If it is a $Z_i$ variable, either in the bivariate estimands $\{A, B\}$ or the conditioning set $\mathbf{C}$, the non-zero samples of the corresponding observational values are used in the test.*
2. *If it is an $R_i$ variable,*
   a) *If it is conditioned (i.e., $R_i \in \mathbf{C}$) and the corresponding true variable is involved (i.e., $Z_i \in \{A, B\} \cup \mathbf{C}$), then it is conditioned by selecting non-zero samples, i.e., $R_i = 0$.*
   b) *Otherwise, all samples of the corresponding binary values of the variable $R_i$ are used.*

Specifically, regarding *2.b)* of Proposition 4, it coincides with a recent proposal to use only binarized counts, though with a different purpose, cell clustering (Qiu, 2020). Note that in our CI estimation, $R_i$ can also be equivalently represented by full samples of the corresponding observation with zeros, $X_i$, without binarization. This is because of the faithfulness in *(A1)*, and the self-masking edges $Z_i \to X_i$, which renders the identical structures, and thus d-separation patterns, among $\mathbf{R}$ and $\mathbf{X}$. However, empirically, it is better to use $\mathbf{X}$, as magnitudes of non-zero values may capture more fine-grained dependencies. E.g., $Z_1 \sim \text{Unif}[-\pi, \pi]$, $Z_2 = \cos Z_1$, and $D_i = \mathbb{1}(R_i < 0)$. Then $X_1 \not\perp\!\!\!\perp X_2$ (faithfulness holds), but this dependence in only captured by magnitudes of $X_1, X_2$ when $R_1 = R_2 = 0$, not by the zero patterns represented as binary indicators, i.e., $R_1 \perp\!\!\!\perp R_2$ (faithfulness violated).

With the essential infrastructure in place for reading off d-separations from data, we are now prepared to develop algorithms to recover the causal structure among $\mathbf{Z} \cup \mathbf{R}$. In addition to discovering the gene regulatory network within $\mathbf{Z}$, our objective extends to uncovering the causes for each $R_i$, i.e., to discover the dropout mechanisms. Since $R_i = D_i$ OR $\mathbb{1}(Z_i = 0)$, an edge $Z_i \to R_j$ exists if and

only if $Z_i \rightarrow D_j$, i.e., gene $i$'s expression influences the dropout of gene $j$. However, we first notice that not all such dropout mechanisms are identifiable. See the natural upper bound in Proposition 2.

Proposition 2 can be readily derived from Proposition 3: to determine the existence of $Z_j \rightarrow R_i$, it is necessary to condition on at least $Z_i$, as otherwise the path $Z_j - Z_i \rightarrow R_i$ is left open. However, since $R_i$ is part of the bivariate estimands, it is impossible to condition on $Z_i$, leading to a contradiction. Consequently, we can only identify edges of the form $Z_i \rightarrow R_j$ where $Z_i$ and $Z_j$ are non-adjacent, meaning there exists a subset $\mathbf{S} \subset [p]$ such that $Z_i \perp\!\!\!\perp Z_j | \mathbf{Z_S}$. The question then arises: since $\mathbf{Z}$ cannot be directed test on, how can we initially identify these non-adjacent pairs based only conditional independencies from the data with dropouts? Interestingly, we have the result in Theorem 3: observed independencies are correct independencies, though correct independencies may not always be observed.

**Theorem 3** (Observed independencies are correct independencies). *Assume (A1), (A2), (A4).* $\forall i, j, \mathbf{S}$,

$$X_i \perp\!\!\!\perp X_j | \mathbf{X_S} \overset{①}{\Rightarrow} X_i \perp\!\!\!\perp X_j | \mathbf{Z_S}, \mathbf{R_S} = \mathbf{0} \overset{②}{\Rightarrow} Z_i \perp\!\!\!\perp Z_j | \mathbf{Z_S}, \mathbf{R_{S \cup \{i,j\}}} = \mathbf{0} \overset{③}{\Rightarrow} Z_i \perp\!\!\!\perp Z_j | \mathbf{Z_S}. \quad \text{(C.1)}$$

Theorem 3 provides insights that align with the previous Proposition 1: although the conditional independencies inferred from observational data with dropouts are "rare", they are accurate. The leftmost hand side of Equation (C.1) represents the direct testing of conditional independence using complete observational data, as mentioned in Proposition 1. The second term is precisely the approach proposed in Theorem 1, involving the removal of samples containing zeros in the conditioning set. The third term is more stringent, as it requires the elimination of samples with zeros in all variables involved in the test, not just the conditioning set. When the structural assumption *(A3)* is in the play, the converse directions of ② and ③ also hold, as stated in Theorem 1. However, in the general setting without *(A3)*, these implications hold only in one direction: the observed independencies must be correct, but the correct independence may not be observed.

For instance, consider the graph $Z_1 \rightarrow Z_2 \rightarrow Z_3$ with $\{Z_i \rightarrow R_i\}_{i=1}^3$, and our focus is on $Z_1 \perp\!\!\!\perp Z_3 | Z_2$. If we introduce the additional edges $Z_1, Z_3 \rightarrow R_2$, it serves as a counterexample to the inverse of ③. Similarly, if we add edges $R_1, R_3 \rightarrow R_2$, it contradicts the inverse of ②. Progressing from left to right, the testing approaches provide increasingly tighter bounds on the correct underlying conditional independence relations, while also in empirical terms, they are less data-efficient due to the increased deletion of samples.

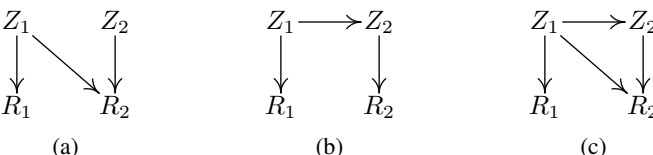

(a)            (b)            (c)

Figure 9: Three exemplar causal dropout graphs that are unidentifiable with each other.

Having established that the influence of dropout mechanisms between adjacent gene pairs in a GRN is unidentifiable, i.e., the presence of an edge $Z_i \rightarrow R_j$ can only be identified when $Z_i$ and $Z_j$ are non-adjacent in the GRN, a subsequent question arises: for a non-adjacent pair of $Z_i$ and $Z_j$ but with $Z_i \rightarrow R_j$, can such a causal effect through dropout mechanisms be distinguished from the causal effect through gene interactions in GRN? The answer is still generally negative. To illustrate this, let us consider the three examples depicted in Figure 9. Cases (b) and (c) are unidentifiable due to the result in Proposition 2, while (b) and (a) are also unidentifiable, as all testable CI relations remain the same, e.g., $R_1 \not\perp\!\!\!\perp R_2$ and $Z_1 \not\perp\!\!\!\perp Z_2 | R_1, R_2$. This finding aligns with the conclusions presented in Theorem 3 and Appendix B.1, namely that the absence of a causal effect, whether mediated through dropout mechanisms or gene interactions, can be ascertained. However, determining the existence and precise nature of a causal effect generally remains elusive.

## C.3    PROOFS OF THE RESULTS IN THE GENERAL CAUSAL DROPOUT MODEL IN §4.2

We first show the proof of Theorem 3, i.e., observed independencies are correct independencies.

*Proof of Theorem 3.* We prove for each implication as follows:

$\overset{①}{\Rightarrow}$: By faithfulness in *(A1)*, $X_i \perp\!\!\!\perp X_j | \mathbf{X_S} \Rightarrow R_i \perp\!\!\!\perp_d R_j | \mathbf{R_S}$ in the compact causal dropout model with only two parts of variables, $\mathbf{Z}$ and $\mathbf{R}$, as shown in Appendix C.2. By definition of d-separation,

either there is no path connecting $R_i$ and $R_j$, or for any such path, there exists a non-collider in $\mathbf{R_S}$, or a collider s.t. it and all its descendants are not in $\mathbf{R_S}$. Now add $\mathbf{Z_S}$ into the conditioning set. For each path, no collider could have itself or its descendants in $\mathbf{Z_S}$, as otherwise it is still already opened by the descendants $\mathbf{R_S}$. Therefore, $R_i \perp\!\!\!\perp_d R_j | \mathbf{Z_S}, \mathbf{R_S}$. By Markov condition in *(A1)*, $X_i \perp\!\!\!\perp X_j | \mathbf{Z_S}, \mathbf{R_S} = \mathbf{0}$.

$\overset{\textcircled{2}}{\Rightarrow}$: Show by its contrapositive. By Markov condition in *(A1)*, $Z_i \not\perp\!\!\!\perp Z_j | \mathbf{Z_S}, \mathbf{R}_{\mathbf{S} \cup \{i,j\}} \Rightarrow Z_i \not\perp\!\!\!\perp_d Z_j | \mathbf{Z_S}, \mathbf{R}_{\mathbf{S} \cup \{i,j\}}$, i.e., there must be a path connecting $Z_i$ and $Z_j$ that is open conditioning on $\mathbf{Z_S}, \mathbf{R}_{\mathbf{S} \cup \{i,j\}}$. 1) If none of such paths has any of $R_i$ or $R_j$ on it, then consider one with two ends extended, i.e., $R_i \leftarrow Z_i \cdots Z_j \rightarrow R_j$: for any non-collider, including $Z_i$ and $Z_j$, it is not in $\mathbf{Z_S}, \mathbf{R_S}$; for any collider, either it or one of its descendants is in $\mathbf{Z_S}, \mathbf{R_S}$ and not in $\{R_i, R_j\}$, as otherwise there must exist another path with $R_i$ or $R_j$ on it, e.g., $Z_i \rightarrow R_i \leftarrow \cdots \leftarrow$ collider $\cdots Z_j$ which is already opened by $R_i$ and rest variables, contradiction. 2) Otherwise, consider a path with $R_i$ or $R_j$ on it. W.l.o.g., consider the segmentation of this path with one end extended, i.e., $R_i \cdots Z_j \rightarrow R_j$. Similarly, it is still open conditioning on $\mathbf{Z_S}, \mathbf{R_S}$. Therefore, we have $R_i \not\perp\!\!\!\perp_d R_j | \mathbf{Z_S}, \mathbf{R_S}$. By faithfulness in *(A1)* and faithful observability in *(A4)*, we have $X_i \not\perp\!\!\!\perp X_j | \mathbf{Z_S}, \mathbf{R_S} = \mathbf{0}$.

$\overset{\textcircled{3}}{\Rightarrow}$: Also show by its contrapositive. By Markov condition in *(A1)*, $Z_i \not\perp\!\!\!\perp Z_j | \mathbf{Z_S} \Rightarrow Z_i \not\perp\!\!\!\perp_d Z_j | \mathbf{Z_S}$. By *(A2)*, $\mathbf{R}_{\mathbf{S} \cup \{i,j\}}$ does not contain any non-collider on the open path connecting $Z_i$ and $Z_j$, and thus involving them into the conditioning set will still keep the paths open, i.e., $Z_i \not\perp\!\!\!\perp_d Z_j | \mathbf{Z_S}, \mathbf{R}_{\mathbf{S} \cup \{i,j\}}$. By faithfulness in *(A1)* and faithful observability in *(A4)*, we have $Z_i \not\perp\!\!\!\perp Z_j | \mathbf{Z_S}, \mathbf{R}_{\mathbf{S} \cup \{i,j\}} = \mathbf{0}$. $\square$

Then, to show the proof of Theorem 2, we will present the sufficient and necessary condition for observing $Z_i \perp\!\!\!\perp Z_j | \mathbf{Z_S}, \mathbf{R}_{\mathbf{S} \cup \{i,j\}} = \mathbf{0}$, and examine the identifiability of the causal effects (either through gene interactions or dropout mechanisms) under the condition.

**Theorem 2** (Identification of GRN and dropout mechanisms). *Assume (A1), (A2), (A4), and (A5). In the CPDAG among $\mathbf{Z}$ output by Definition 2, if $Z_i$ and $Z_j$ are non-adjacent, it implies that they are indeed non-adjacent in the underlying GRN, and for the respective dropout mechanisms, $Z_i$ does not cause $R_j$ and $Z_j$ does not cause $R_i$. On the other hand, if $Z_i$ and $Z_j$ are adjacent, then in the underlying GRN $Z_i$ and $Z_j$ are non-adjacent only in one particular case, and for dropout mechanisms, the existence of the causal relationships $Z_i \rightarrow R_j$ and $Z_j \rightarrow R_i$ must be naturally unidentifiable.*

*Proof of Theorem 2.* We have shown a necessary condition for observing non-adjacent $Z_i, Z_j$ as in Theorem 3, i.e., they are truly non-adjacent in GRN. Now we give the sufficient condition (stronger): for at least one $\mathbf{Z_S}$ that d-separates $Z_i$ and $Z_j$ in the GRN, for any variable involved, i.e., $\forall k \in \mathbf{S} \cup \{i,j\}$, $R_k$ is neither a common children of $Z_i, Z_j$, nor a descendant of a common children of $Z_i, Z_j$ (if any). In other words, at least one conditional independence between $Z_i, Z_j$ needs to be observed.

We show this by contrapositive: if $Z_i, Z_j$ are truly non-adjacent in GRN but for all $\mathbf{Z_S}$ that d-separates $Z_i$ and $Z_j$ in the GRN, the corresponding conditional independencies cannot be observed, i.e., $Z_i \perp\!\!\!\perp_d Z_j | \mathbf{Z_S}$ but $Z_i \not\perp\!\!\!\perp_d Z_j | \mathbf{Z_S}, \mathbf{R}_{\mathbf{S} \cup \{i,j\}}$, then for each of such $\mathbf{S}$, there must be an open path connecting $Z_i, Z_j$ in the form of $Z_i \rightarrow W \leftarrow Z_j$, where either $W \in \mathbf{R}_{\mathbf{S} \cup \{i,j\}}$, or there exists a $k \in \mathbf{S} \cup \{i,j\}$ s.t. $W$ is $R_k$'s ancestor. Otherwise, if the open path has more than one variable $W$, then at least one variable is not a collider and can be conditioned on to block the path, contradiction.

Therefore, if $Z_i$ and $Z_j$ are non-adjacent in the output by Definition 2, they must be truly non-adjacent and does not cause each other's dropout. Otherwise if $Z_i$ and $Z_j$ are adjacent in the output by Definition 2, they are either truly adjacent, or in a non-adjacent but particular case as shown above. In either case, the existences of $Z_i \rightarrow R_j$ and $Z_j \rightarrow R_i$ are unidentifiable, as illustrated in Proposition 2 and Figure 9(a).

$\square$

# D SUPPLEMENTARY EXPERIMENTAL DETAILS AND RESULTS

## D.1 LINEAR SEM SIMULATED DATA

Here we provide a more detailed elaboration on the simulation study and implementation as in §5.1.

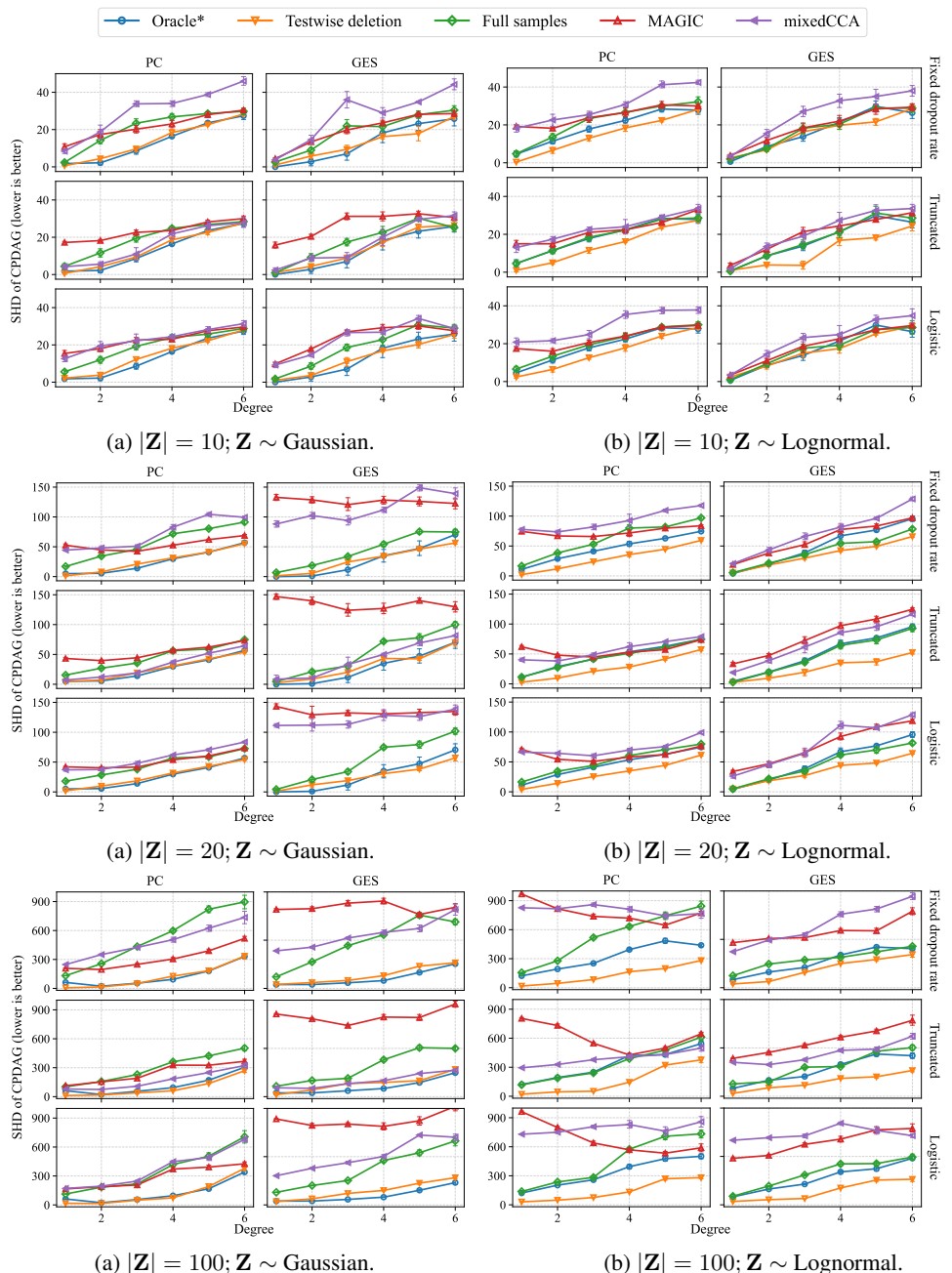

Figure 10: Experimental results (SHDs of CPDAGs) of 10, 20, and 100 variables on simulated data.

**Methods** We investigate the effectiveness of our method (denoted as testwise deletion method) by comparing it to other methods on simulated data. Specifically, we consider the following baselines: (1) MAGIC (Van Dijk et al., 2018) that imputes the dropouts, and (2) mixedCCA (Yoon et al., 2020) that applies mixed canonical correlation to estimate the correlation matrix of $\mathbf{Z}$. The imputed data and estimated correlation matrix by MAGIC and mixedCCA, respectively, can be used as input to a suitable causal discovery algorithm. Furthermore, we also consider another baseline that directly applies a causal discovery algorithm to full samples (without deleting or processing the dropouts), which correspond to the method in Proposition 1. We also report the results assuming that the complete data of $\mathbf{Z}$ are available, denoted as Oracle*, in order to examine the performance gap between these methods and the best case (but rather impossible) scenarios without dropouts. We

consider PC Spirtes et al. (2000) and GES (Chickering, 2002) as the causal discovery algorithms for these methods. We describe the hyperparameters of these methods in detail in Appendix D.1.

**Simulation setup**  We randomly generate ground truth causal structures based on the Erdös–Rényi model (Erdős et al., 1960) with $p \in \{10, 20, 30\}$ nodes and average degree of 1 to 6. On each setting five causal structures are sampled, corresponding to the confidence intervals in Figure 4. For each causal structure, we simulate 10000 samples for the variables $\mathbf{Z}$ according to a linear SEM, where the nonzero entries of weighted adjacency matrix are sampled uniformly from $(-1, -0.25) \cup (0.25, 1)$, and the additive noises follow Gaussian distributions with means and standard deviations sampled uniformly from $(-1, 1)$ and $(1, 2)$, respectively. In this case, $\mathbf{Z}$ are jointly Gaussian; we also consider another case by applying an exponential transformation to each variable $Z_i$, such that $\mathbf{Z}$ follow Lognormal distributions. Lastly, we apply three different types of dropout mechanisms on $\mathbf{Z}$ to simulate $\mathbf{X}$: (1) dropout with the fixed rates, (2) truncating low expressions to zero, and (3) dropout probabilistically determined by expression, which are described in Examples 1, 2, and 3, respectively. For the first two categories of dropout mechanisms, we randomly set 30% to 70% of samples on each gene to get dropped out. For the third mechanism, we set $D_i \sim \text{Bernoulli}(\text{logit}(-1.5Z_i - 0.5))$.

**PC**  We use the implementation from the `causal-learn` package[3] (Zheng et al., 2023). For speed consideration, we use FisherZ as the conditional independence test and set the significance level `alpha` to 0.05. Note that theoretically FisherZ cannot be used, as after the log transformation, dropout, or zero-deletion distortion, the data is generally not joint Gaussian. However, since the result is already good, we did not further use the much slower but asymptotically consistent Kernel-based conditional independence test (Zhang et al., 2011). It is also interesting to see why FisherZ still empirically works here, and examine under which condition the FisherZ test is still asymptotically consistent even after test-wise deletion (one trivial case is Definition 2 where $\mathbf{Z}$ follows joint Gaussian and dropout happens with fixed rates). We leave this as an interesting future work.

**GES**  We use the Python implementation[4]. We modify the local score change by first calculating the test statistics of the corresponding testwise-deletion CI relation, and then rescale the partial correlation so that it produces the same test statistics in the full sample case. This partial correlation and the full sample size are then used to calculate the BIC score change. The $L0$ penalty is set to 1.

**Experimental results for Figure 4**  The results of 30 nodes are provided in Figure 4. It is observed that our proposed method (1) has a much lower SHD compared to the baselines across most settings, especially when PC is used as the causal discovery algorithm, and (2) leads to SHD close to the oracle performance. Furthermore, one also observes that directly applying causal discovery to full samples results in a poor performance in most cases, possibly because, as indicated by Proposition 1, the estimated structures by this method may contain many false discoveries, leading to a low precision. It is worth noting that mixedCCA is specifically developed for jointly Gaussian variables $\mathbf{Z}$ and truncated dropout mechanism. Therefore, compared to the other settings, it performs relatively well in this setting, despite still having a higher SHD than our proposed method. This further validates the effectiveness of our method that outperforms existing parametric models with no model misspecification. One may wonder why test-wise deletion method outperforms Oracle* in the Lognormal setting. A possible reason is that, after zero deletion, many small values (especially for truncated and logistic dropout mechanisms) are removed; therefore, the resulting distribution is closer to Gaussian, which is more suitable for the specific CI test adopted for these methods.

**Additional result 1: On different scales**  As a complement to Figure 4 (30 nodes), here we also present the results of SHDs of CPDAGs in different scales (small as 10 and 20 nodes, and large as 100 nodes), as shown in Figure 10. Impressively, our proposed test-wise zero deletion still performs best among baselines across various settings.

Moreover, to better echo the high dimensionality in real-world scRNA-seq data (e.g., around 20k genes for humans), we also conduct experiments in a super large scale: The ground-truth graph has 20,000 nodes and an average degree of 5. The sample size is 20,000. The dropout rate is 70%. We used the FGES (Ramsey et al., 2017) implementation[5] on an Apple M1 Max with 10 cores. Here are the results:

---

[3]https://github.com/py-why/causal-learn/blob/main/causallearn/search/ConstraintBased/PC.py
[4]https://github.com/juangamella/ges
[5]https://github.com/cmu-phil/tetrad

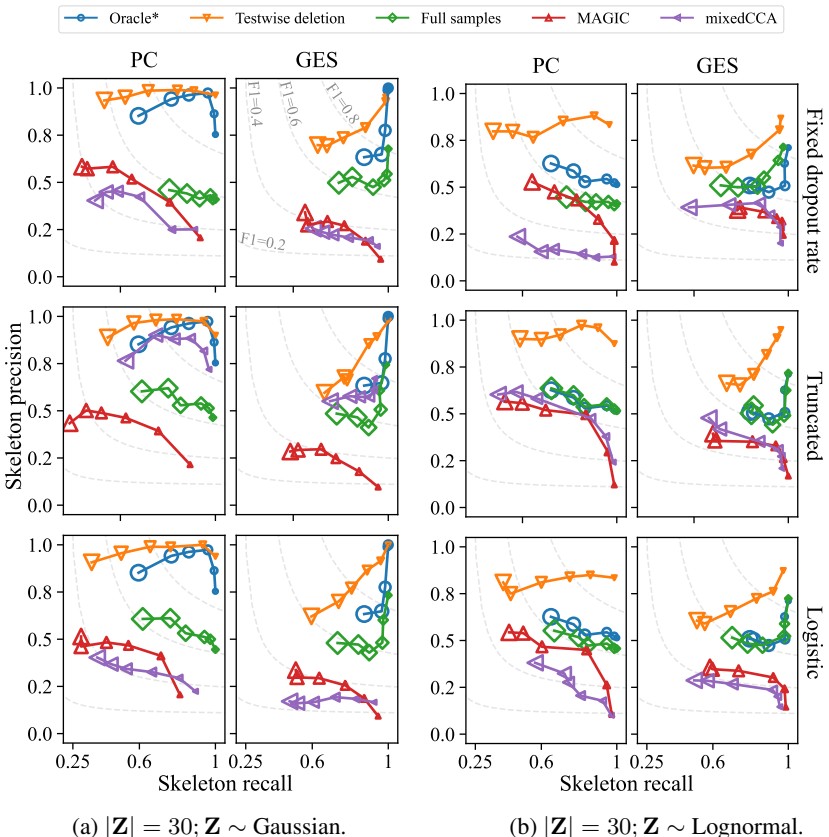

Figure 11: Experimental results of 30 variables on simulated data (Figure 4) using another metric: precisions and recalls of skeleton edges. Each subplot corresponds to a specific algorithm (PC or GES) on a specific base data distribution (Gaussian or Lognormal) with a specific dropout mechanism (Fixed dropout rate, Truncated, or Logistic). In each subplot, the 5 lines correspond to the 5 dropout-handling strategies (Oracle*, Testwise deletion, Full samples, MAGIC, or mixedCCA). On each line, the 6 markers with sizes from small to big correspond to the average graph degrees from 1 to 6.

Table 1: FGES results on a large-scale graph with 20,000 nodes.

|                    | Oracle* | 0del  | Full samples* |
|--------------------|---------|-------|---------------|
| Skeleton F1 score  | 0.93    | 0.90  | 0.14          |
| Time               | 1421s   | 1512s | > 24 hrs      |

As FGES directly on samples with dropouts has not ended within 24 hrs (since there are too many false dependencies), we reported the F1-score (0.14) given by the graph at the last iteration before we terminate it. We see that: 1) Zero-deletion is effective in recovering the true structures, and 2) Zero-deletion is fast, because it corrects the false dependencies and the graph in search is sparser than that on dropout-contaminated data.

Regarding the speed (particularly in the high dimensionality), it is worth noting that our proposed zero-deletion method itself is a straightforward technique that introduces no additional time complexity. As a simple and versatile tool, it can be seamlessly integrated into various other algorithms, and the whole time complexity is completely contingent on the specific algorithm it accompanies (e.g., PC, GES, or FGES, or GRNI-specific methods in §5.2).

**Additional result 2: On different metrics than SHD**    As SHD metrics can sometimes be misleading, we also calculate another metric, precision and recall of skeleton edges, on the simulation setting

| Dropout rate % | 0 | 10 | | 30 | | 50 | | 70 | | 90 | |
|---|---|---|---|---|---|---|---|---|---|---|---|
| Method | Full | Full | 0del | Full | 0del | Full | 0del | Full | 0del | Full | 0del |
| Skel Precis | .98 | .58 | **.98** | .54 | **.98** | .57 | **.98** | .54 | **1.0** | .44 | **1.0** |
| Skel Recall | 1.0 | **.98** | .98 | **.96** | .89 | **.98** | .89 | **.98** | .60 | **.82** | .16 |
| Skel F1 | .99 | .73 | **.98** | .69 | **.93** | .72 | **.93** | .70 | **.75** | .57 | 27 |
| PDAG SHD | 19 | 56 | **23** | 62 | **28** | 55 | **28** | 61 | **28** | 81 | **43** |

(a)

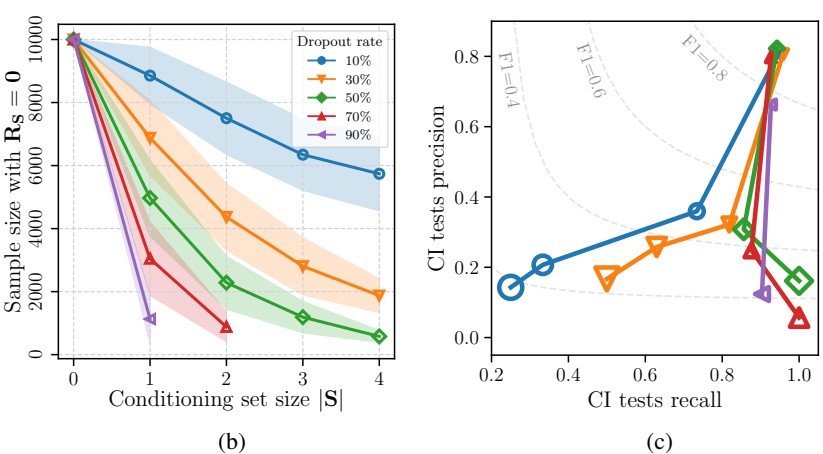

(b) (c)

Figure 12: Experimental results to test the validity of assumption *(A5)* (effective sample size after test-wise deletion) with varying dropouts. On a dataset simulated with 30 variables and average graph degree of 3, truncated dropout mechanism is applied to produce 5 datasets with dropout rates = 10%, 30%, 50%, 70%, and 90%, respectively, shown by the 5 colors above. PC with test-wise deletion is run on each dataset. **(a)**: The structure identification accuracies w/wo test-wise deletion; **(b)**: The remaining sample size after test-wise zero deletion, for each CI test during the PC run, with conditioning set sizes $|\mathbf{S}|$ growing from 0 to a maximum of 4. The marker and the band show the mean and the standard deviation of sample sizes under a specific size $|\mathbf{S}|$; **(c)**: The precisions and recalls of all CI tests (d-separation / conditional independencies are counted as positives) during the PC run, where each colored line is on the data with a specific dropout rate, and each marker on the line is with a specific conditioning size $|\mathbf{S}|$. Marker sizes from small to big correspond to the conditioning size $|\mathbf{S}|$ from 0 to the bigger.

of Figure 4 (because PC and GES output edges without strengths, we can only calculate the precision and recall without thresholding curves). The results are shown in Figure 11.

The results strongly validate our method's efficiency: we see from Figure 11 that the orange lines (Testwise deletion) are often above all other lines, i.e., achieving consistently highest precisions. This aligns with our Proposition 1 and Theorem 1: testwise deletion can correctly recover conditional independencies, i.e., reduce false dependencies, and thus reduce false positive estimated edges.

**Additional result 3: On varying dropout rates**   While in §4 we demonstrate the validity of assumption *(A5)* (effective sample size after testwise deletion) in a real-world examplar dataset (Figure 3), here we also conduct a set of synthetic experiments with varying dropout rates to test the validity of assumption *(A5)*. On a dataset simulated with 30 variables and average graph degree of 3, we examined the truncated dropout mechanism with dropout rates varying from 10% to 90%.

Figure 12 depicts the results of the remaining sample sizes in PC runs, the accuracies of CI tests, and the overall structure identification accuracies. These results support the validity of assumption *(A5)*: even when the dropout rate is as high as 70%, the remaining samples sizes for CI tests are still considerably high to avoid Type-II errors, leading to consistently better SHDs and skeleton F1-scores.

## D.2 Realistic Synthetic and Curated Data under the BEELINE Framework

Except for the linear SEM experiments, we also conduct synthetic experiments that are more 'realistic' biological setting (closer to real scRNA-seq data) and more 'competitive' (on algorithms specifically desgined for GRNI, not just PC and GES). Specifically, we use the BEELINE GRNI benchmark framework (Pratapa et al., 2020). For data simulation, we use the BoolODE simulator that basically simulates gene expressions with pseudotime indices from Boolean regulatory models. We simulate all the 6 synthetic and 4 literature-curated datasets in (Pratapa et al., 2020), each with 5,000 cells and a 50% dropout rate, following all the default hyper-parameters. For algorithms, in addition to PC and GES, we have attempted to encompass all 12 algorithms benchmarked in (Pratapa et al., 2020). However, due to technical issues, five of them were not executable, so we focused on the other seven ones: SINCERITIES, SCRIBE, PPCOR, PIDC, LEAP, GRNBOOST2, and SCODE. For each algorithm, five different dropout-handling strategies are assessed, namely, oracle*, testwise deletion, full samples, imputed, and binarization (Qiu, 2020).

Our test-wise zero-deletion strategy, as a simple and versatile tool, is employed in these SOTA algorithms as follows. We divided the seven SOTA algorithms into three categories: 1) For algorithms that explicitly compute (time-lagged) partial correlations, including SCRIBE, PPCOR, and PIDC, the test-wise zero-deletion can be readily plugged in as in PC and GES; 2) For algorithms that entail gene subset analysis (e.g., regression among gene subsets) but without explicit partial information, including SINCERITIES and LEAP, in each subset analysis we delete samples containing zeros in corresponding genes; and 3) For algorithms that estimate the structure globally without any explicit subset components, including GRNBOOST2 and SCODE, we directly input data with list-wise deletion, i.e., delete samples containing zeros in any gene.

Note that the evaluation setting is not in favor of our implementation (especially for PC and GES with testwise deletion), due to the following three reasons: 1) Model mis-specification for PC and GES, both of which assume i.i.d. samples while BoolODE simulated cells are heterogeneous with pseudo-time ordering. These pseudotime indices are input to the SOTA algorithms as additional information, when applicable; 2) Model mis-specification for the CI tests. For speed consideration, in PC and GES we simply use FisherZ and BIC score which assume joint Gaussianity, instead of non-parametric CI tests and generalized scores; and 3) Other 7 SOTA algorithms output edges with strengths so we report the best F1-score through thresholding, while PC and GES do not output strengths.

Even though, the results still strongly affirm the efficacy of our method's efficacy, as supported by the F1-scores of the estimated skeleton edges in Figure 5. See analysis in §5.2. Note that in Figure 5, the minimum of the colorbar is anchored to 0.5 in order to visualize the differences among the majority of entries with a finer resolution – among all the 450 entries there are 380 ones with values $\geq 0.5$. F1 values $<0.5$ yields a relatively poor performance and lacks significant reference value.

Notably, one may wonder why there is a seemingly "performance degeneration" from full data to zero deletion for PC algorithms in the four curated BEELINE datasets. In fact, this performance degeneration is more likely attributed to the PC algorithm itself, rather than the zero-deletion method. Among the 4 curated datasets, there is a performance degeneration on two of them: HSC and GSD. Notably however, it is exactly on these two datasets where full-samples performs even better than oracle*, i.e., dropouts even do "good", instead of harm. This contradiction with theory makes the further analysis of zero-deletion less meaningful on these 2 cases. While on the remaining 2 cases (F1-oracle*$\geq$F1-full samples), exactly we also have F1-0del$\geq$F1-full samples. We further extend this insight across all cases:

- Among the 65 dataset-algorithm pairs where dropouts indeed do harm (F1-full sample$<$F1-oracle*), zero-deletion yields efficacy in dealing with dropouts (F1-0del$>$F1-full sample) on 47 (86.2%) of them.
- Among the 17 dataset-algorithm pairs where dropouts unreasonably do good (F1-full sample$>$F1-oracle*), zero-deletion yields efficacy (F1-0del$>$F1-full sample) on only 4 (23.5%) of them.

Interestingly, the above statistics reaffirms the efficacy of zero-deletion (Theorem 1): zero-deletion yields CI estimations (and thus GRNI results) more accurate/faithful to oracle* – regardless of whether the oracle* is better or worse than full samples. As for the reasons why in some cases dropouts unreasonably performs even better, we will investigate more into details later.

### D.3 REAL-WORLD EXPERIMENTAL DATA

We conduct experiments on three representative real-world single-cell gene expression datasets to verify the effectiveness of our proposed method, including data from Bone Marrow-derived Dendritic Cells (BMDC) (Dixit et al., 2016), Human Embryonic Stem Cells (HESC) (Chu et al., 2016) and Chronic Myeloid Leukemia Cells (CMLC) (Adamson et al., 2016). The three datasets consist of 9843, 758, and 24263 observational samples respectively. Following the previous exploration of genetic relations, we determine the "ground-truth" GRN using the known human gene interactions from the TRRUST database (Han et al., 2018).

For the BMDC (Dixit et al., 2016) data, we follow the practice in (Dixit et al., 2016; Saeed et al., 2020) and only consider the subgraph on 21 transcription factors believed to be driving differentiation in this tissue type. As is shown in Figure 6, using the deletion based CI test as in Definition 1, the PC estimated graph is much sparser, i.e., more true conditional independencies are identified, as shown by the red edges. Furthermore, more known interactions are identified, shown by the blue edges. Note that for PC's output, for several undirected edges in CPDAG, we manually orient them in the favored direction from known interactions.

Table 2: SHD Results of PC on HESC and CMLC datasets, using different methods as in Figure 4.

| Subgraph | Testwise deletion | Full samples | MAGIC |
|---|---|---|---|
| HESC | | | |
| 12 nodes, 9 edges | **16** | 20 | 23 |
| 16 nodes, 13 edges | **20** | 26 | 23 |
| 16 nodes, 11 edges | **19** | 30 | 25 |
| 18 nodes, 17 edges | **26** | 38 | 29 |
| 19 nodes, 16 edges | **25** | 34 | 28 |
| CLMC | | | |
| 16 nodes, 14 edges | **28** | 49 | 41 |
| 17 nodes, 15 edges | **33** | 65 | 46 |
| 18 nodes, 15 edges | **32** | 70 | 47 |
| 21 nodes, 16 edges | **41** | 98 | 42 |
| 23 nodes, 18 edges | **33** | 106 | 53 |

For the rest two datasets HESC (Chu et al., 2016) and CMLC (Adamson et al., 2016), to enable a more comprehensive experimental comparison while also considering the computation efficiency of causal discovery algorithms, we randomly we randomly sample five subgraphs on each dataset to evaluate the performances of different methods. The subgraphs are sampled with the following criteria: no loops, no hidden confounders, and each gene should have a zero sample size less than 500 for HESC, or a zero-rate less than 70% for CMLC. The last condition is for the test power of deletion based CI tests: if a gene have too many zeros across all cells, then for any CI relation conditioned on it, the sample size after removing the zero samples will be too small, leading to an empirical bias. Thus we do not consider those genes. This is reasonable as if a gene is almost always zero across all cells (either not expressed or all dropped out), it may not be that important or actively involved in the genes interaction in these cells. On these sub-datasets, we conduct PC on full observational samples, on test-wise deleted samples, and on MAGIC imputed data, respectively. The SHD results are shown in Table 2. We notice that the graphs produced by `Testwise deletion` have the smallest SHDs, i.e., closest to the groundtruths. One may further be curious why the SHDs are generally larger than the number of edges in ground-truth, i.e., even an empty graph would have a smaller SHD. This is because the graph discovered from data is usually much denser than the groundtruths, i.e., there may be additional gene interactions that are yet not known or not recorded in the TRRUST database (Han et al., 2018). To this end, we also examine the number of true positive edges (i.e., the detected true interactions) returned by each method, where `Testwise deletion` is slightly better than `Full samples`, and significantly better than `MAGIC`.

