# OpenReview forum: "Gene Regulatory Network Inference in the Presence of Dropouts: a Causal View"
_ICLR.cc/2024/Conference — ICLR 2024 oral_

### Official Review · Reviewer_6osj · 2023-10-29

**Soundness:** 3 good
**Presentation:** 4 excellent
**Contribution:** 4 excellent
**Rating:** 8
**Confidence:** 4

**Summary:**

The authors tackle the task of inferring gene regulatory networks (GRNs) from single-cell RNA sequencing (*scRNA-seq*) data, which is made difficult by the fact that *scRNA-seq* data showcases many zero values, which are either due to technical reasons (dropouts) or biological (no gene expression). This missing data problem can lead to overly-dense graphs being produced by SOTA causal discovery algorithms applied to this task. The authors propose a `Causal Dropout Model` to characterize the dropout mechanism and come up with a simple solution to handle dropouts, namely conditioning on non-zero entries in the conditioning set when performing conditional independence sets. They show that this approach is sound under relatively mild assumptions and performs well on a large number of synthetic, semi-synthetic, and real-world data sets.

**Strengths:**

The problem tackled is very important, as gene regulatory networks can provide direct insight into the workings of biological mechanisms, and *scRNA-seq* is increasingly available for this task. The authors present their idea in an almost flawless manner, with sufficient attention being given to describing related work, providing relevant examples, and to framing and testing the assumptions made for the `Causal Dropout Model`. Finally, the authors showcase the performance of their approach in an extensive series of experiments, in which they examine different dropout mechanisms, different causal discovery and GRN inference-specific algorithms, on multiple settings and types of data.

**Weaknesses:**

My only (minor) gripes are that most of the theoretical analysis is deferred to the appendix, which can sometimes lead to questions due to insufficient detail (see below), and that the references are a bit sloppy.

Miscellaneous comments:
- page 4, Example 4: I think the Bernoulli distributions should be reversed, since there should be a minus sign in the denominator exponential when computing the logistic function.
- page 10, reference typo: * after "Tabula Sapiens Consortium"
- page 11, formatting error: "ALBERTS" should not be capitalized
- page 11, there seems to be no reference to Gao et al. (2022) in the paper.
- page 12, duplicate author: "Paul R. Rosenbaum"

**Questions:**

1. On page 6, when discussing the connection between BIC score and Fisher-Z test statistics, how do you conclude that "only local consistency is actually needed"? The reference in Nandy et al. (2018) shows this connection, but does not say anything about the assumptions made in the GES paper. Could you elaborate on this point?
2. On page 7, what do you mean by "identifiability upper bound" for identifying the dropout mechanisms? How is this upper bound characterized?
3. Perhaps I missed it, but in Theorem 2, what is the "one particular case" in which Z_i and Z_j are non-adjacent in the underlying GRN.
4. In Figure 4(b), how do you explain that testwise deletion performs better than the oracle? Shouldn't that be the best case scenario?
5. I haven't seen *causal sufficiency* mentioned anywhere, which is an important assumption made when using algorithms like PC and GES. Is it reasonable for this type of data to assume that there are no hidden confounders? In either case, I would say something about this important practical limitation.

---

> ### Author Response · Authors · 2023-11-16
> **Response to Reviewer 6osj**
>
> We are grateful for the reviewer's positive feedback and constructive comments. Please see below for our response.
>
> ---
>
> **(Q1)** The reviewer requests clarification on the claim that "(for the correctness of GES), only local consistency of the scoring function is actually needed".
>
> **R:** Thank you for your feedback on this point. We have revised this part in the [updated submission](https://openreview.net/pdf?id=gFR4QwK53h) to make the logical connection clearer. Here is a summary: first, the fact that only local consistency is required for GES's consistency can already be seen from Chickering (2002)'s proof. Specifically, their Lemma 9 and Theorem 4 (main results) are grounded on local edge modifications. Then, we mention Nandy et al. (2018) as they make this point more explicit: they derive the BIC score change directly as the partial correlation involved in local consistency definition (Lemma 5.1), and further go beyond BIC score, suggesting "defining generalized scoring criterions by... conditional independence". This is recognized and implemented by [4] at the same time, and later by [5], [6].
>
> ---
>
> **(Q2)** The reviewer wonders for some technical details regarding the identifiability result without assumption A3 (i.e., when dropout can be affected by other genes). Specifically,
>
> > "What is the 'identifiability upper bound' for identifying the dropout mechanisms?"
>
> **R:** The identifiability upper bound of dropout mechanisms is characterized by Proposition 2 in Appendix C.1: "If $Z_i$ and $Z_j$ are truly adjacent in GRN, then whether they affect each other's dropout, i.e., whether edges $Z_i \rightarrow R_j$ and $Z_j \rightarrow R_i$ exist, is naturally unidentifiable". This is because in this case, the (in)dependence between e.g., $Z_j$ and $R_i$ given at least $Z_i$, is untestable (see Proposition 3, Testability of CI queries).
>
> Then, what if $Z_i$ and $Z_j$ are truly non-adjacent in GRN? Theorem 2 indicates that this non-adjacency, together with the absence of $Z_i \rightarrow R_j$ and $Z_j \rightarrow R_i$, can be identified barring one particular case. So,
>
> > "What is this 'one particular case'"?
>
> **R:** This particular case is when the non-adjacent $Z_i$ and $Z_j$ **always share a common child** (not descendant -- so we say it as so *particular*) that has to be conditioned on. This is formally defined in Proof of Theorem 2: "For every $\mathbf{S} \subset \mathbf{Z}$ that d-separates $Z_i$ from $Z_j$, there is an open path $Z_i\rightarrow W \leftarrow Z_j$, where either $W \in \mathbf{R}_{\mathbf{S}\cup \{i,j\}}$, or there exists a $k \in \mathbf{S}\cup \{i,j\}$ s.t. $W$ is $R_k$'s ancestor". Only in this particular case, the true d-separation cannot be recovered, resulting in a false adjacency.
>
> We thank you for your detailed interest in these aspects of our work. For further clarity, we hope you will find Appendix C, particularly the illustrative examples (Figures 8 and 9), helpful.
>
> ---
>
> **(Q3)** The reviewer wonders why test-wise deletion performs better than the oracle in Figure 4(b).
>
> **R:**  As is discussed in Appendix D.1, regarding why test-wise deletion method outperforms the oracle in the lognormal setting, one possible explanation is that, after zero deletion, many small values (especially for truncated and logistic dropout mechanisms) are removed; therefore, the resulting distribution is closer to Gaussian. Given that our experiments use Fisher's Z tests for conditional independence -- merely for speed consideration, as discussed in our Q3 response to reviewer pofH -- this closer alignment with Gaussian distribution may inadvertently enhance the CI test accuracies.
>
> ---
>
> **(Q4)** The reviewer suggests explicitly mentioning the causal sufficiency assumption, given its significance in the context of constraint-based methods.
>
> **R:** We appreciate your emphasis on this. Indeed, causal sufficiency was implicitly included here in our assumption A1, as one of the "common assumptions needed for the asymptotic consistency of constraint-based methods". We have made it explicit in the [updated submission](https://openreview.net/pdf?id=gFR4QwK53h). Furthermore, we completely agree that it is an important future research problem -- note that the proposed zero deletion for CI correction is general and can be seamlessly incorporated into any existing constraint-based methods, and it would be interesting to see how this idea can be incorporated into existing local causal discovery methods, or methods that allow hidden confounders.

---

> ### Author Response · Authors · 2023-11-16
> **Response to Reviewer 6osj (cont'd)**
>
> **(Q5)** Several miscellaneous comments, including
>
> - **Bernoulli distribution in Example 4:** We appreciate your careful examination. Please kindly note that upon double check, we believe the current one is indeed correct: the term in Bernoulli(·), e.g., logistic$(\log 2 - Z_i)$, is the probability of dropout ($D_i = 1$), which is why the density function of $Z_i$ is multiplied not by this term, but by one minus it, i.e., logistic$(-(\log 2 - Z_i))$. We have replaced the name "logistic" with "sigmoid" in the [updated submission](https://openreview.net/pdf?id=gFR4QwK53h) for clarity.
> - **Formatting errors in references:**  We thank you for pointing out these errors. We have thoroughly reviewed the references and corrected the formatting issues as much as possible in the [updated submission](https://openreview.net/pdf?id=gFR4QwK53h).
> -   **Theoretical analysis deferred to appendix:**  We are keeping improving the way to deliver our results comprehensively. Our current structuring was due to page limit and the relatively dense contents in this paper -- we chose to focus the main body on key messages, illustrative examples, and experimental highlights. Many theoretical details, though very interesting (e.g., testability of A3), had thus to be deferred to appendix.
>
> Thank you again for your encouraging and valuable input!
>
> ---
>
> [4] Huang, Biwei, et al. "Generalized score functions for causal discovery." _Proceedings of the 24th ACM SIGKDD international conference on knowledge discovery & data mining_. 2018.
>
> [5] Chickering, Max. "Statistically efficient greedy equivalence search." _Conference on Uncertainty in Artificial Intelligence_. PMLR, 2020.
>
> [6] Shen, Xinwei, et al. "Reframed GES with a neural conditional dependence measure." _Conference on Uncertainty in Artificial Intelligence_. PMLR, 2022.

---

> > ### Comment · Reviewer_6osj · 2023-11-21
> > **Acknowledgment**
> >
> > Thank you for the detailed rebuttal and for clarifying the doubts I had.

---

### Official Review · Reviewer_pofH · 2023-10-30

**Soundness:** 3 good
**Presentation:** 3 good
**Contribution:** 3 good
**Rating:** 8
**Confidence:** 4

**Summary:**

The authors propose a method for causal discovery of gene regulatory networks (GRNs), i.e., causal gene-gene relationships. Instead of conducting conditional-independence tests on the raw data without consideration of the dropout patterns of single cell sequencing (e.g., doing PC algorithm on the entire data), they propose conducting the tests only on non-zero conditioning variables, by showing that the conditional independence relations of variables conditioned on variables with non-zero values are the same as for data without dropouts. The authors demonstrate that their method outperforms competing state-of-the-art methods in causal discovery of GRNs on several data sets.

**Strengths:**

- The method is in my opinion very original and will certainly be of actual use in the field of computational biology.
- The paper is clearly written and easy to follow. As far as I can judge, the authors demonstrate an excellent grasp of the contemporary literature, both in causal discovery as well as in computational biology.
- The method outperforms state-of-the-art methods for GRN inference in several benchmarks.
- The experimental section is convincing, and should be easy to reproduce.

**Weaknesses:**

I do not have major comments on possible weaknesses.

**Questions:**

- The authors state "while a zero entry of $X_2$ may be noisy (i.e., may be technical), a non-zero entry of $X_2$ must be accurate, i.e., biological.". As far as I can tell, non-zero values might also be technical due to sequencing/mapping/algorithmic errors, correct?
- Supposing elevated dropout rates of, say 50%, the method relies on the fact that a conditioning variable can be found which "breaks" dependencies. Is this correct?
- Using Fisher's $z$-test assumes multivariate Gaussianity. Wouldn't a kernel-based independence test be better for the log-normal data?
- Figure 5 is not very readable. I believe a simple table or something similar could improve the presentation.
- The source code could be improved and properly documented (What libraries are required? Which versions of the libraries did you use for validation? How do I run all experiments? Etc etc.).

---

> ### Author Response · Authors · 2023-11-16
> **Response to Reviewer pofH**
>
> We sincerely appreciate the reviewer's encouragement and insightful feedback. Please see below for our response.
>
> ---
> **(Q1)** The reviewer wonders whether non-zero values could also be "technical", i.e., subject to sequencing errors.
>
> **R:** Thank you for this insightful point. As discussed in Appendix B.1, it is indeed possible that non-zero values are also influenced by technical errors, and that should be addressed separately from the general measurement error issue [1]. In this paper, we focus only on the dropout errors, mainly for the following two reasons:
>
>  - Unlike general measurement errors that are ubiquitous in empirical sciences with a long history of research, dropout errors are **peculiar to scRNA-seq data** (in contrast to bulk RNA-seq data), owing to specific technical challenges like low RNA capture efficiency (Section 1). Our study thus focuses on this more specific and recent issue in the field.
>  - Empirically, dropout errors in scRNA-seq data tend to be **more harmful** to downstream tasks than measurement errors [2]. Non-zero values, despite potential technical errors, are usually centered around the true expression levels, with errors manifested as relatively minor random noise. It is less likely for a true zero to be falsely reverse-transcribed into a non-zero cDNA count (out of no mRNA molecules). However, a true non-zero can be dropped out to zero, and this happens excessively, leading to a substantial distortion of the true gene expressions. This distortion is a more pressing concern, and thus forms the crux of this research focus.
>
> ---
> **(Q2)** The reviewer wonders whether the method relies on the fact that non-zero values of conditioned variables can be found, even with an elevated dropout rate of, say 50%.
>
> **R:** Yes, our method relies on it, as exactly outlined in assumption A5, which states that a sufficient sample size of conditioned variables remains after zero deletion. A5 is justified in the following aspects:
>
>  - A5 is commonly satisfied in real scRNA-seq data, thanks to the inherent sparsity of GRNs. As a gene usually only interacts with a relatively small number of other genes, we usually don't need to condition on a large number of variables, resulting in larger remaining sample sizes. This is empirically supported by our **real data experiments (Figure 3)**.
>  - A5 is also validated by a synthetic experiment with varying dropout rates (Figure 12 in Appendix D.1): even when the dropout rate is **as high as 70%**, the remaining sample sizes for CI tests are still considerably high to avoid large Type-II errors, leading to consistently better SHDs and skeleton F1-scores.
>  - Additionally, it's worth highlighting the advantages of scRNA-seq technologies which can nowadays easily produce hundreds of thousands of samples within a single run. This innovation renders concerns over sample size somewhat mitigated.
>
> ---
> **(Q3)** The reviewer suggests using kernel-based CI tests for log-normal data, instead of the Fisher's Z tests currently used.
>
> **R:** We appreciate your suggestion. Our use of Fisher's Z tests (and BIC scores) was merely for speed consideration in experimental runs. Although there is empirical evidence suggesting the approximate suitability of Pearson partial correlations in Gaussian copula models [3], and our results indeed improve a lot over baselines (that's why we didn't further include the result by more accurate but slower conditional independence testing methods, such as KCI-test), we can't agree more that kernel-based tests, theoretically, would be more appropriate here -- not only for log-normal setting but also for Gaussian setting (both (a) and (b) in Figure 4), as after the dropout distortion, the joint Gaussianity among $\mathbf{Z}$ is also not preserved in $\mathbf{X}$ anymore, even with zero deletions.
>
>
> ---
> **(Q4)** The reviewer recommends improving the clarity of presentations, such as in Figure 5 and in code documentation.
>
> **R:** Thanks for raising this suggestion. In response, we have revised the explanation for Figure 5 in our [updated submission](https://openreview.net/pdf?id=gFR4QwK53h) for better clarity. Additionally, we have enriched the documentation in the [updated supplementary codes](https://openreview.net/attachment?id=gFR4QwK53h&name=supplementary_material) to provide more detail.
>
> Thank you again for your positive feedback and valuable suggestions!
>
> ---
>
> [1] Cochran, William G. "Errors of measurement in statistics." _Technometrics_ 10.4 (1968): 637-666.
>
> [2] Sarkar, Abhishek, and Matthew Stephens. "Separating measurement and expression models clarifies confusion in single-cell RNA sequencing analysis." _Nature genetics_ 53.6 (2021): 770-777.
>
> [3] Kim, Jong-Min, et al. "Partial correlation with copula modeling." _Computational statistics & data analysis_ 55.3 (2011): 1357-1366.

---

### Official Review · Reviewer_f2vv · 2023-10-31

**Soundness:** 3 good
**Presentation:** 3 good
**Contribution:** 3 good
**Rating:** 6
**Confidence:** 4

**Summary:**

The authors proposed a causal graphical model, named causal dropout model, to characterize the dropout mechanism in scRNA-seq data.
They found that simply ignore the data points in which the conditioned variables have zero values can still lead to consistent estimation of conditional independence (CI) relations with those in the original data.

**Strengths:**

The task of inferring gene regulatory network is of interest especially in the bioinformatic domain.
The empirical results seem to indicate that the approach can be integrated into existing causal discovery methods to handle dropouts.
Writing and presentation skill is well.

**Weaknesses:**

For network inference, they can use some evaluation metrics such as ROC curve or PR curve to assess how well their predicted network recovers the true network.
They should conduct more experiments to show the performance gained by using their causal dropout model.
Several gene network inference methods have been designed to handle missing values in scRNA-seq data. Therefore, as a practical analytical framework, the authors should prioritize the comparison of their model with the most advanced existing network inference methods.

**Questions:**

For network inference, they can use some evaluation metrics such as ROC curve or PR curve to assess how well their predicted network recovers the true network.
They should conduct more experiments to show the performance gained by using their causal dropout model.
Several gene network inference methods have been designed to handle missing values in scRNA-seq data. Therefore, as a practical analytical framework, the authors should prioritize the comparison of their model with the most advanced existing network inference methods.

---

> ### Author Response · Authors · 2023-11-16
> **Response to Reviewer f2vv**
>
> We sincerely appreciate the reviewer's constructive comments and helpful feedback. Please see below for our response.
>
> ---
> **(Q1)** The reviewer suggests examining metrics like ROC curves.
>
> **R:** Thanks for the suggestion! We have indeed incorporated these metrics in the experiments of our experiments. Please see below for details:
>
>  - In **Figure 11 of Appendix D.1**, we show the precisions and recalls in skeleton edges identification, and the results strongly validate our method's efficiency: Test-wise deletion performs best with consistently highest precisions. This aligns with our Proposition 1 and Theorem 1: test-wise deletion correctly recovers conditional independencies, i.e., reduces false dependencies (edges).
>  - In **Figure 5 of Section 5.2**, we investigate our proposed zero-deletion approach on all realistic synthetic and curated datasets, with PC, GES, and all SOTA algorithms in the BEELINE framework. The SOTA algorithms output edges with strengths so the best F1-scores through thresholding are reported, while since PC and GES do not output strengths, only the results from a fixed significance level are reported. Even so, the efficacy of the proposed zero-deletion (on both PC, GES, and on other SOTAs) are clearly demonstrated.
>
>
> ---
> **(Q2)** The reviewer suggests conducting more experiments and comparing with advanced existing network inference methods.
>
> **R:** We thank the reviewer for the suggestion, and would like to note that we have indeed comprehensively evaluated our method with the SOTA GRNI methods using the BEELINE benchmarking framework. Please refer to **Section 5.2** for details. Below is a summary:
>
>  - For datasets, we follow a more realistic simulation setup (BoolODE), and evaluate the methods on **all the 10 synthetic and curated datasets** benchmarked in BEELINE.
>  - For approaches, we examine **PC, GES, and all the 7 SOTA algorithms** (if executable, including SINCERITIES, SCRIBE, PPCOR, PIDC, LEAP, GRNBOOST2, and SCODE) benchmarked in BEELINE.
>  - For dropout-handling strategies, we comprehensively examine **5 representative methods**, 'oracle', 'zero-deletion', 'full samples', 'imputed', and 'binarized', where the last two are most commonly used in the current literature.
>
> The experimental results, as depicted in **Figure 5**, clearly demonstrate that the proposed zero-deletion is effective in dealing with dropouts, with consistent benefits across different integrated algorithms and on different datasets.
>
> ---
>
> Once again, we are grateful for the reviewer's valuable comments, and sincerely hope that you will find your suggestions and concerns well addressed by the response above and the relevant sections in the paper. Your further feedback would be appreciated, and we hope for the opportunity to respond to it.

---

> ### Author Response · Authors · 2023-11-22
> **Could you please let us know whether our responses properly addressed your suggestions and concerns?**
>
> Dear Reviewer f2vv,
>
> Thank you very much for your time spent on our submission. We have tried to address your suggestions and concerns, particularly regarding the evaluation metrics and the comparison with existing GRNI methods. These aspects, as you rightly pointed out, are crucial for demonstrating the efficacy of our approach. And we would like to highlight that **indeed, these aspects have already been included/addressed in our submission.**
>
> Could you please let us know whether your concerns were properly addressed? Given that the discussion involving authors will end in a day, if there are any other concerns or questions that need further clarification, please let us know, and we will immediately respond to them.
>
> If you find your suggestions and concerns well addressed by the response above and the relevant sections in the submission, we would greatly appreciate any re-assessment of our work in light of them. We recognize the demands on your time and deeply value your continued engagement and feedback on our submission. Thank you.
>
> Best wishes, Authors of submission 489

---

### Author Response · Authors · 2023-11-22
**Global response to reviewers and AC**

We thank the reviewers for their valuable feedback and constructive comments.

We are encouraged by their assessment of the problem tackled as "very important" (Reviewer 6osj) and "of interest especially in the bioinformatic domain" (Reviewer f2vv), our method as "very original", "certainly be of actual use" (Reviewer pofH), and "can be integrated into existing methods" (Reviewer f2vv), our presentation as "almost flawless", "clearly written" and "easy to follow" with "an excellent grasp of the literature" (all the reviewers), and our empirical performance as "outperforming" and "convincing" through "an extensive series of experiments" (Reviewers pofH, 6osj).

---

**Main changes to the updated submission:**

We have [updated our submission](https://openreview.net/pdf?id=gFR4QwK53h) following the reviewers' suggestions, with all the changes highlighted in blue. Below is a summary:

+ We have revised the explanation for Figure 5 (experimental results on BEELINE benchmark) for better clarity.
+ We have enriched the documentation in the updated supplementary codes to provide more details to reproduce our results.
+ We have revised the paragraph regarding why GES only requires local consistency to make the logical connection clearer.
+ We have made the assumption A1 more explicit, including causal sufficiency.
+ We have thoroughly reviewed the references and corrected the formatting issues as much as possible.

For details and additional miscellaneous changes, please kindly refer to the individual responses.

---

Thank the reviewers again for the valuable input. We are happy for any further questions/feedback during the discussion period.

Best wishes, Authors of submission 489

---

### Meta-Review · Area_Chair_LFSG · 2023-12-05

**Metareview:**

The authors analyze a causal model that allows them to handle (basically by ignoring them) the positivity violations in the data, specifically for gene regulatory networks. The reviewers found the writing of the paper clear, the idea interesting and original, and with potential impact. Experimental results were found convincing.

**Justification For Why Not Higher Score:**

N/A

**Justification For Why Not Lower Score:**

The importance of the problem and creative solution with a special focus on an impactful application area make this a valuable contribution worthy of being highlighted as an oral talk.

---

### Decision · Program_Chairs · 2024-01-16

Accept (oral)